# Identification of electro-mechanical interactions in wind turbines

Fiona D. Lüdecke[1], Martin Schmid[2], and Po Wen Cheng[1]

[1]Stuttgart Wind Energy at Institute of Aircraft Design, Universtity of Stuttgart, Allmandring 5B, 70569 Stuttgart, Germany
[2]Insitute of Electrical Energy Conversion, University of Stuttgart, Pfaffenwaldring 47, 70569 Stuttgart, Germany

**Correspondence:** Fiona D. Lüdecke (luedecke@ifb.uni-stuttgart.de)

**Abstract.** Large direct drive wind turbines with a multi-megawatt power rating face design challenges, especially concerning tower top mass, due to scaling laws for high-torque generators. This work proposes to extend the design space by moving towards a more system-oriented approach, considering electro-mechanical interactions. This requires an extension of the state-of-the-art wind turbine models with additional degrees of freedom. To limit the computational effort of such models, a profound understanding of possible interaction mechanisms is required. This work aims to identify interactions of an additional degree of freedom in the radial direction of the generator with the wind turbine structure, the aerodynamics and the wind turbine controller. Therefore, a Simpack model of the IEA 15MW RWT is implemented and coupled to a quasi-static analytical generator model for electromagnetic forces. The analytical model, sourced from literature, is code-to-code validated against a finite element model of the generator in Comsol Multiphysics. Electro-mechanical simulation results do not show interactions with the aerodynamics or the controller. However, interactions with the wind turbine structure occur. It is shown that the modelling approach can affect the system's natural frequencies, which can potentially impact the overall system design choices.

## 1 Introduction

The increasing contribution of wind energy to the energy transition is based on two major aspects: more capacity being installed and the development of new wind turbines with increasing rated power. Recently, manufactures have reached up to 16 MW rated power for offshore wind turbines (WTs). These new releases use geared and direct-drive concepts. Both concepts go along with design challenges. In the past, gearboxes statistically showed a high failure rate, leading to high maintenance needs (Reder et al., 2016). The direct-drive concepts are connected to large low-speed generators with a diameter of about 10 m (Gijs van Kuik, 2016), leading to a large component mass. Generally, generator mass can be divided into active mass, contributing to power production, and passive mass, ensuring structural support only. Scaling laws show that upscaling existing generator designs increases the passive mass over-proportionally compared to the active mass (Shrestha et al., 2009). In consequence, the design requirements for tower and foundation raise to ensure that the tower top mass is carried safely.

This work focuses on direct-drive concepts: As a reaction to the scaling laws of generators, research has started to investigate diverse approaches to reduce the mass of these large generators. These efforts include design optimisation algorithms (Delli Colli et al., 2012; Tartt et al., 2021), the investigation of new manufacturing techniques as additive manufacturing (Hayes et al., 2018) and the usage of alternative generator concepts (Mueller and McDonald, 2009).

The key requirement for generator optimisation is to ensure sufficient clearance between stator and rotor, the air gap, at all

times. This mainly depends on the structural stiffness of the design and the electromagnetic forces. The electromagnetic forces can be divided into radial and tangential forces acting on the surface of the rotor and stator. The tangential forces create the generator torque that is required for power production, while the radial forces do not contribute to power production. When the rotor and stator are perfectly centred to each other, the radial forces are balanced in all directions. This is a purely theoretical case, as unequal magnetisation of magnets and small deviations in the design do not allow for a perfect alignment in practice. Furthermore, direct-drive generators in wind turbines are subject to highly variable excitation compared to other power plant applications. This means, besides variations of rotational speed, also fluctuating bending moments on the shaft occur, which push the generator into eccentricity. The electromagnetic forces $F_{\mathrm{mag}}$ increase non-linearly with decreasing air gap length $\delta$, according to $F_{\mathrm{mag}} \sim \frac{1}{\delta^2}$. As a result, unbalanced electromagnetic forces cause or increase eccentricity.

In consequence, common design requirements call for high structural rigidity for the generator design and the main bearings to avoid eccentricity (Hayes et al., 2018). These requirements result from optimisation at the isolated component design level. However, WTs are complex systems with strong interactions between the physical phenomena involved. It has been shown that a more system-oriented design approach using multidisciplinary design, analysis and optimisation (MDAO) techniques is required (Dykes et al., 2011).

For generator design, an electro-mechanical model is needed that couples the mechanical and the electromagnetic system behaviour of the generator. To set up such a model, the questions of the required model fidelity and how the models are connected have to be answered, as outlined in Perez-Moreno et al. (2016). Multiple research projects have developed electro-mechanical generator models. Most of them focus on the component level (Boy and Hetzler, 2019; Hayes et al., 2018; Duda et al., 2019) or look only into the torsional coupling of aerodynamic and generator torque along the shaft as torsional spring-damper (Novakovic et al., 2013).

Coupled models, including a full WT, were introduced by Sethuraman et al. (2017) and Cardaun et al. (2021). Cardaun et al. (2021) use a look-up table for the electromagnetic forces at each magnetic pole that depends on the local air gap length. The look-up table was generated using a finite element (FE) model of one pole pair and running a high number of FE simulations with varying air gap length. The resulting WT model is used for acoustic analyses. Sethuraman et al. (2017) use an analytical model from Sethuraman et al. (2014) to describe the generator electromagnetic forces per pole. The investigation is divided into the analysis of electro-mechanical interactions in an onshore turbine and the analysis of drive-train loads for a floating offshore WT. Based on the comparison of controller signals using the onshore turbine, the study concludes that feedback from the drive-train to other turbine components can be neglected.

In summary, the electro-mechanical investigations in literature focus on specific WT designs and try to optimise them. A study to identify the mechanisms of interactions has not been found. To extend the design space in the future, it is expected that a profound understanding of the boundary conditions under which interactions can occur is required. Therefore, this work aims to investigate the physical mechanisms behind electro-mechanical interactions in WTs and identify those with the potential to influence the WT behaviour on a system level. A coupled model of WT and generator is introduced. Specifically, a radial degree of freedom (DoF) at the generator is included, while the axial DoF along the drive-train shaft and tilting DoFs of the drive-train are omitted. The high impact of aerodynamic damping in fore-aft direction is expected to lead to a reduced relevance

of the axial DoF for electro-mechanical interactions. Additionally, the axial displacement only reduces the effective core length of the generator. In the expected range of millimeters and less, this is will have a minor impact on the radial generator forces. Based on the results in Duda et al. (2019) tilting is expected to influence occurring load levels in the bearings. Potential impacts on the wind turbine dynamics, though, can not be accurately predetermined. However, combining several DoFs will increase the complexity of the interactions. Therefore, it was decided to concentrate only on the radial DoF and its implications on the system response, to maintain clarity and coherence on the scope of the paper. Nevertheless, the extrapolation of the results in this work to the axial and tilting DoFs are discussed in Sec. 3.1. With the additional radial DoF, the interactions with the WT structure, the aerodynamics and the controller are analysed. Furthermore, the effects of the added DoF and the electromagnetic generator forces are investigated to distinguish between the two.

The paper is structured as follows: The models used for the analysis are introduced in Sect. 2. Sect. 3 discusses the interactions in the WT system by dividing them into structural interactions, interactions with the aerodynamics and interactions stemming from the WT controller behaviour. Sect. 4 provides a short conclusion of the work and outlines potential future work.

## 2 Modelling

In this work, the analyses of electro-mechanical interactions in WTs are based on numerical simulations. These simulations require a description of the WT's components and the physical phenomena involved. This has to include the WT blades' structure and the aerodynamics, the structural properties of the tower, the controller for operating the turbine, and the drive-train with the subcomponents, i.e. the shaft, the bearings and the generator.

Modelling all the aforementioned components can lead to high computational effort. Therefore, available state-of-the-art WT models simplify the drive-train to a torsional spring-damper and exclude the electromagnetic characteristics of the generator. This simplification can not be followed when analysing electro-mechanical interactions due to eccentricity, on a turbine level. Therefore, the state-of-the-art models have to be extended to a coupled model of WT and generator. The details about the models used in this work and their differences to the state-of-the-art are explained in this section, starting with the WT model in Sect. 2.1 and followed by the generator model in Sect. 2.2.

### 2.1 Wind turbine

The WT model used in this work is based on the IEA 15 MW reference WT defined by the IEA Wind Task 37 as reported in Gaertner et al. (2020). It includes a drive-train tilt of 5 deg and blade cone angles of 2.5 deg in upwind configuration. As WT controller, the recommended ROSCO controller (NREL, 2021) is used with the wind speed estimator set to option 1, namely the Immersion and Invariance Estimator. All other options equal the default settings of the IEA 15 MW RWT OpenFAST model, using a constant torque control above rated and the TSR tracking PI-controller for below rated conditions, and a second-order low-pass filter for generator speed and pitch control signals. Hydrodynamics around the monopile have not been considered.

To enable the analysis of electro-mechanical interactions, this work requires adding a DoF. In OpenFAST, this is connected to high coding efforts. Thus, the model is transferred to Simpack, a multi-body (MB) simulation software that is widely used

**Table 1.** Natural frequencies of the isolated components and of the coupled system of the wind turbine model in Simpack in baseline configuration, i.e. in equivalent implementation to OpenFAST according to Gaertner et al. (2020)

| Mode | Isolated in Hz | Coupled system in Hz |
| --- | --- | --- |
| Blade flap 1 | 0.54 | 0.55 |
| Blade edge 1 | 0.73 | 0.73 |
| Blade flap 2 | 1.60 | 1.61 |
| Tower FA 1 | 0.78 | 0.19 |
| Tower SS 1 | 0.78 | 0.19 |
| Tower FA 2 | 3.31 | 1.23 |
| Tower SS 2 | 3.31 | 1.30 |
| Monopile FA 1 | 5.12 | 4.00 |
| Monopile SS 1 | 5.12 | 4.18 |
| Monopile tor 1 | 18.93 | 5.10 |
| Monopile FA 2 | 23.39 | 11.25 |
| Monopile SS 2 | 23.39 | 14.98 |
| Monopile axial 1 | 30.08 | 10.23 |

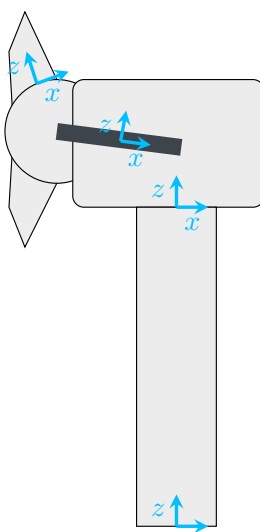

**Figure 1.** Definition of coordinate systems at the tower base, the tower top, the generator centre and the blade root. All coordinate systems are right-handed coordinate systems with the y-axis pointing into the plane. At the generator centre, a stationary and a rotating coordinate system are defined.

for WT applications and offers a higher flexibility for the definition of DoFs avoiding additional coding efforts (Simpack). The
95 modelling approach is equivalent to the implementation in OpenFAST, following the coordinate system definition according
to Fig. 1 and using flexible blades, tower and substructure. The resulting natural frequencies of the isolated components in
one-sided clamping are provided in Tab. 1 together with those system natural frequencies of the coupled system, for which
the according mode is predominant. The drive-train shaft has been modelled as a rigid component due to its high diameter to
length ratio.

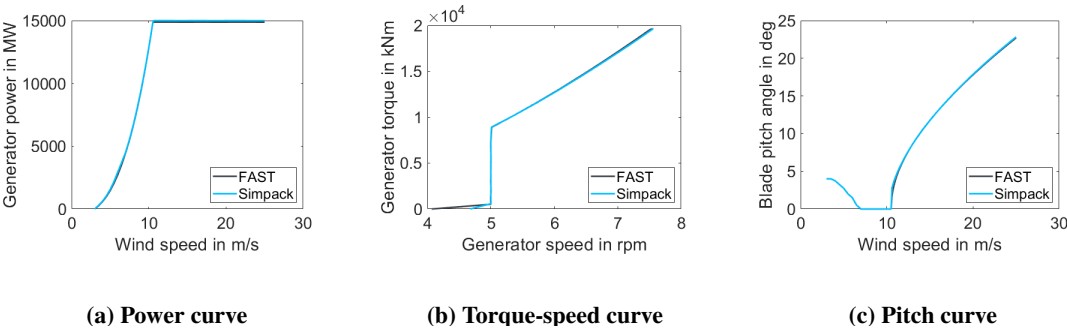

(a) Power curve            (b) Torque-speed curve            (c) Pitch curve

**Figure 2.** Comparison of steady state behaviour of the implemented Simpack model in baseline configuration against the OpenFAST reference model described in Gaertner et al. (2020).

The resulting Simpack model was validated against OpenFAST. The comparison of the steady states of both models are
given in Fig. 2. From left to right, the power curve, the torque-speed curve and the pitch curve are shown. All the curves show
a very good agreement. The comparison of the dynamic behaviour of the two models is based on a stepped wind field, which
is shown in Fig. 3. This comparison confirms the agreement of both models under dynamic loading over the full operational
range.

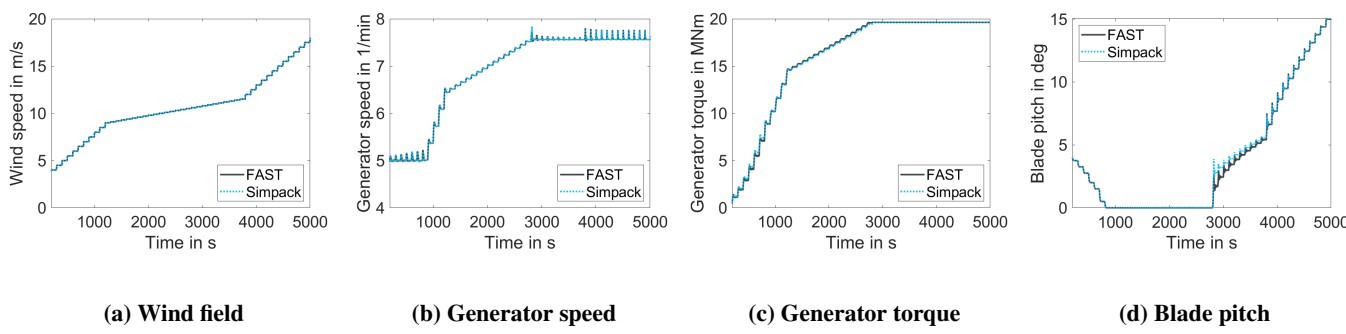

(a) Wind field       (b) Generator speed       (c) Generator torque       (d) Blade pitch

**Figure 3.** Comparison of dynamic system behaviour of the implemented Simpack model in baseline configuration against the OpenFAST reference model described in Gaertner et al. (2020).

The implementation of the additional DoF is explained in details in Sect. 2.1.1, also outlining the motivation for this model adaptation. The Simpack solver settings used throughout the study are summarised in Sect. 2.1.2.

### 2.1.1 Model extensions

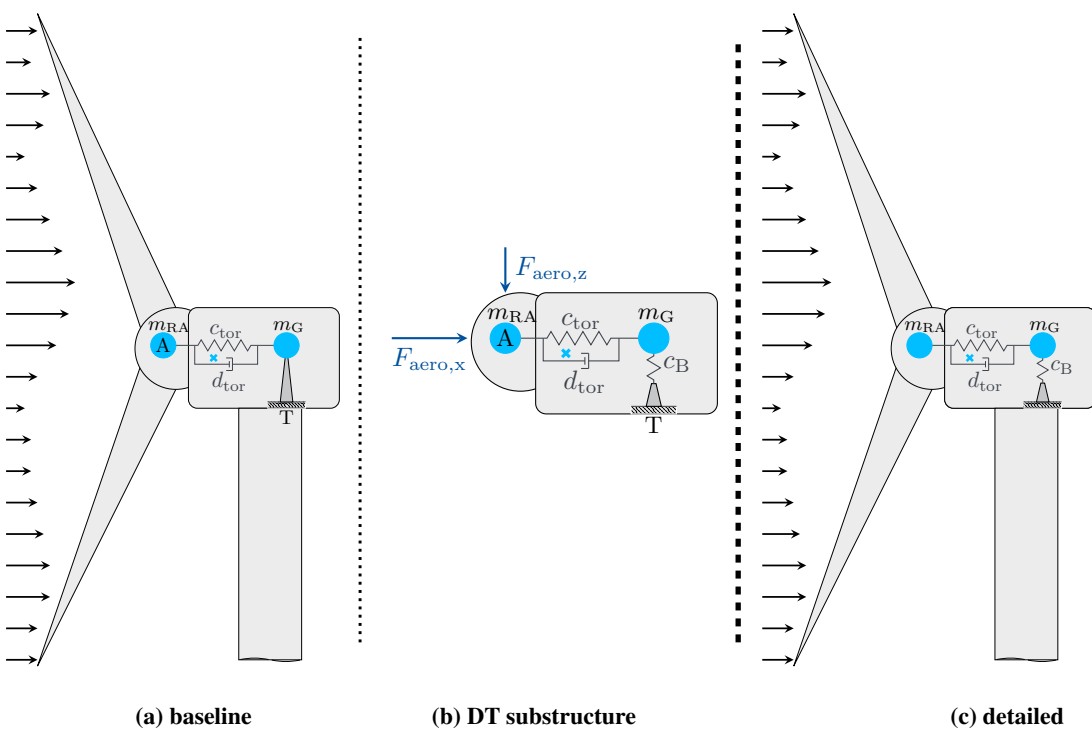

**(a) baseline**        **(b) DT substructure**        **(c) detailed**

**Figure 4.** Modelling approaches for WT load analysis (a) in baseline configuration according to the state-of-the-art providing the input for (b) the substructure model of the detailed drive-train in comparison to (c) the model introduced in this work including the detailed drive-train into the full WT model.

     A sketch of the state-of-the-art modelling approach for a complete WT is given in Fig. 4 (a). This model is referred to as the baseline model. The foundation is omitted in the figure. The drive-train is modelled as a rotating point mass $m_G$ with a

mass moment of inertia around its rotation axis. This point mass is connected to the WT rotor with a torsional spring damper element, representing the torsional stiffness $c_{tor}$ and damping $d_{tor}$ of the main shaft. Between the rotating and non-rotating components in the nacelle, only the rotational DoF is enabled. The mass $m_G$ can only rotate around the shaft axis, and translational displacements into any direction are not allowed due to the rigid support.

     The non-rotating components are also reduced to a point mass with a mass moment of inertia around the three axes of the

nacelle. This point mass includes all non-rotating tower top components, e.g. the nacelle housing and power electronic devices. The resulting centre of gravity of the nacelle, including rotating and non-rotating drive-train components, is marked with the little, light-blue x.

Electro-mechanical interactions result from non-uniform air gaps in the generator, which can be caused by different effects. This study focuses on eccentricity due to radial displacements. The introduced baseline model though is based on the state-of-the-art assumption of a perfectly aligned generator rotor and stator. Therefore, radial DoFs, allowing for eccentricity, are not included into the baseline WT model.

Past investigations on electro-mechanical interactions, therefore, mainly used a two-step approach, which assumes that the dynamics of the electro-mechancial interactions do not affect components outside the drive-train. In this approach, first, the forces and dynamics of the WT are calculated with the model in Fig. 4 (a). The resulting forces in the hub centre, marked with an **A**, together with the tower top movements at point **T**, are then used as input to a detailed drive-train model without blades and tower as illustrated in Fig. 4 (b). The detailed drive-train model includes the additional DoF at the location of the generator in the radial direction. This requires the description of the main bearing support. In the figure, this is represented by the changed support at the generator location between (a) and (b).

With this work, the assumption that electro-mechanical interactions do not affect components outside the drive-train is questioned, and it is aimed to identify general mechanisms of interactions. Thus, to analyse the impact of the electro-mechanical interactions on the tower and blades, the detailed drive-train model has to be integrated into the full WT model, as shown in Fig. 4 (c). This way a one-step approach for drive-train load calculation can be achieved.

The bearings can be modelled with varying levels of detail. The simplest representation would be a radial spring. More detailed representations would include the rolling elements as rigid bodies, resulting in an MB model of the bearing. The highest level of detail could be achieved using the FE model. With the level of detail, the possibilities of analysing load distributions and dynamic effects inside the bearing increase. At the same time, the computational effort of solving the model grows significantly. For this study, only the supporting behaviour of the bearings is relevant and detailed analyses of the bearing's internal load distribution are not considered. Therefore, the simplest representation as a radial spring is deemed sufficient, minimising the computational effort.

The spring can be modelled by two approaches: a linear and a non-linear stiffness curve. The first is characterised by a constant stiffness value $c$, while the latter uses a displacement-dependent stiffness curve. Commonly, it is assumed sufficient for electro-mechanical interaction analyses to model a bearing as a linear spring, with its spring stiffness constant $c$ in $\frac{\text{N}}{\text{m}}$ (Jaen-Sola and McDonald, 2014; Nejad et al., 2019; Sethuraman et al., 2014; Boy and Hetzler, 2019). The WT documentation lists as main bearings a fixed front bearing and a floating back bearing. Displacements along the shaft axis only decrease the effective length of the generator, which has minor effects to electro-mechanical interactions. Thus, the axial DoF is excluded in this work, to reduce computational costs. Therefore, the floating back bearing is reduced to a fixed bearing.

A more detailed representation of the drive-train of the IEA 15 MW RWT is shown in Fig. 5. Subfigure (a) illustrates the position of the two bearings according to the design in the report (Gaertner et al., 2020). The resulting implementation of the DoFs in Simpack is shown in subfigure (b). The shaft is rigid and thus, one axial constraint at the generator centre is sufficient and avoids over constraining the system. In consequence, the bearings can be reduced to their supporting behaviour, which is indicated by the radial springs in subfigure (b). Additionally, tilting of the generator is not included in this study, which is indicated by the groove in the figure.

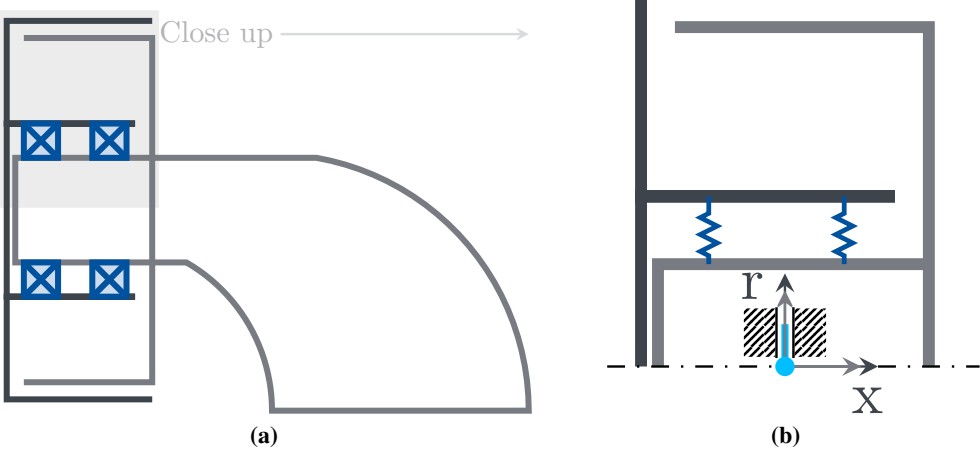

**Figure 5.** Schematic representation of the drivetrain design with outer rotor (dark grey) and inner stator (grey) (a) showing the bearing positions (dark blue) along the shaft and (b) a close up, showing the implementation of the DoFs at the generator between rotor and stator (light blue) and the bearing forces as linear springs (dark blue). Only radial movements are possible, tilting and axial movements are constraint.

For both bearings, a stiffness value has to be specified. State-of-the-art methods model the bearing as an FE model, based on its geometry, to derive the stiffness values. The exact specifications of materials and bearing components, e.g. rolling element geometry, are not available for the investigated WT design. Therefore, the stiffness constants of the given bearing configuration can not be derived. However, the bearing stiffness (BS) is required for the model and is estimated in the following:

The static loading due to gravity, which the bearings have to carry, consists of the mass of the WT rotor assembly and the generator rotor. This mass is used to derive a first estimation of the BS. The WT's rotor assembly weighs 274.9 t and the generator rotor 151.8 t. This means that the two bearings have to carry together 4.186 MN. The maximum allowed radial eccentricity of the generator according to the design is 2 mm (Gaertner et al., 2020). To the knowledge of the authors, no references about common bearing stiffnesses in wind turbines or eccentricity due to loading exist. Therefore, the assumption is made that the gravity loading should only cause a maximum of 10 % of the allowed eccentricity, i.e. 0.2 mm. Thus, the two bearings require an effective BS of 20.93 GN/m. Due to the modelling approach of a rigid shaft with only a radial DoF the distribution of the effective bearing stiffness to the two equivalent force elements does not influence the system behaviour. For simplicity, the bearing stiffness is distributed equally to the two bearings. Therefore, a value of 10 GN/m each is assumed throughout this work. This value represents a first estimation for this study and needs further investigation if a realistic load analysis is intended with the model. Nevertheless, it will serve the purpose of analysing the interaction mechanisms, which are expected to be independent of the exact value of the bearing stiffness.

The radial DoF adds a natural frequency $f_B$ to the system depending on the rotating mass $m_{rot}$, being the sum of the rotor assembly and the generator rotor, and the chosen BS $c_B$ per bearing. In case of an equivalent one-mass-spring-system, the frequency can be calculated to $f_{B,est} = \frac{1}{2\pi} \sqrt{\frac{2 \cdot c_B}{m_{rot}}} = 34.5\,\text{Hz}$, with $m_{rot} = 426.7\,\text{t}$ and $c_B = 10\,\text{GN/m}$.

### 2.1.2 Solver settings

Besides the definition of DoFs or parameters of the WT components like masses and stiffnesses, the parameters of the numerical solver have to be also specified. Specifically, the communication intervals between the modules of aerodynamics and structural solver, and to the controller are determined. Furthermore, the general tolerance criteria are set.

Due to the added system frequency at 34.5 Hz, the communication of the structural solver with the aerodynamic solver has to be chosen accordingly to avoid aliasing effects. A standard approach assumes that the minimum communication frequency to choose is twice the maximum system frequency (Nyquist criteria). To ensure that aliasing is avoided, a higher factor is recommended. Preliminary studies showed a high sensitivity of the results to the chosen communication interval. Details are explained in Sect. 3.2. The results presented in this work use a communication interval to the aerodynamic solver of 0.0001 s. The controller was initially called every 0.02 s in accordance with Gaertner et al. (2020). This interval showed resonance behaviour for the model with resolved drive-train. The resonances were caused by the natural frequency of the added DoF, as this frequency is above the controller's Nyquist frequency of 25 Hz. Therefore, the communication interval was decreased to 0.007 s increasing the Nyquist frequency of the controller to 70 Hz.

The solver tolerances in Simpack were set to default values of $\Delta s_{\mathrm{abs},i} = 10^{-7}$ for the absolute tolerance and $\Delta s_{\mathrm{rel},i} = 10^{-5}$ for the relative tolerance. Only the tolerance for positions has been adapted to $\Delta s_{\mathrm{abs,pos}} = 10^{-9}$ m for absolute tolerance. This results from the expected magnitude of generator eccentricity. The upper bound results from the allowable eccentricity of 2 mm. The lower bound is driven by the generator size of 10 m diameter. An assembly tolerance in the range of nanometres for such a large component is expected to cause unreasonably high cost. At the same time, the maximum allowable eccentricity should be reserved for extreme cases and not occur in normal operation. Therefore, the expected mean eccentricity is assumed to be in the range of $\mu$m. Excluding the electromagnetic forces, the range of occurring eccentricity lowers further. To ensure trustworthy results also for the calculated eccentricity, the chosen tolerance has to be significantly smaller. Defining the tolerance for positions at $10^{-9}$m – a factor of thousand smaller than the assumed mean range of eccentricity of $\mu$m $= 10^{-6}$m – is expected to be sufficient.

## 2.2 Generator

The generator used in this work is related to the IEA 15 MW RWT. An overview of the outer-rotor generator design is given in Tab. 2. More details about the definition of the generator can be found in Gaertner et al. (2020). Two models of different fidelity are investigated, an analytical model (Sect. 2.2.1) and an FE model called numerical model in the following (Sect. 2.2.2). The interactions with the wind turbine focus on the radial generator forces. Variations of the generator torque due to torque ripple or due to the eccentricity are not considered. Those variations mainly apply to the torsional DoF, for which interactions have been investigated intensively in the literature, e.g. (Novakovic et al., 2013). Instead, the state-of-the-art approach is used for the generator torque, based on a look-up table of generator torque over rotational speed. Focusing on the radial forces only allows limiting the computational effort and better isolate the impact of the radial variations to the wind turbine loading. However, the

**Table 2.** Overview of generator parameter for IEA 15 MW RWT according to Gaertner et al. (2020)

| Parameter | Value | Unit |
|---|---|---|
| Air gap radius | 5.08 | m |
| Pole number | 200 | - |
| Slot number | 240 | - |
| Air gap length | 10 | mm |
| Core length | 2.17 | m |
| Rated torque | 21.03 | MNm |
| Rated speed | 0.792 | $\frac{\text{rad}}{\text{s}}$ |
| Electrical frequency | 12.6 | Hz |
| Stator windings per phase | 2 | - |
| Nominal winding current | 4525.48 | A |
| Remanent flux density $B_\text{r}$ | 1.28 | T |
| Relative permeability $\mu_\text{r}$ | 1.06 | - |

remaining computational effort is still significant. Therefore, only the analytical model is used in coupled simulations with the
WT. Prior to the coupled simulations, the numerical model is used in Sect. 2.2.3 to verify the analytical model.

### 2.2.1   Analytical model

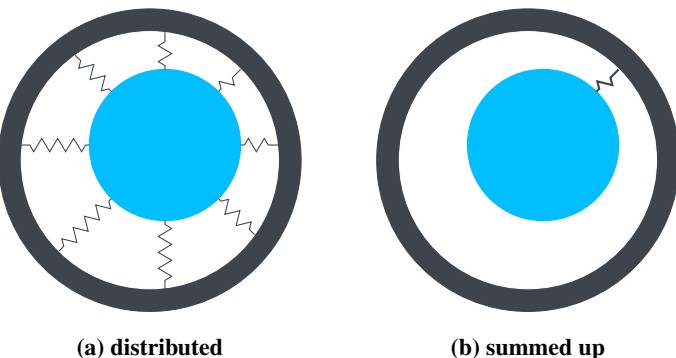

**(a) distributed**          **(b) summed up**

**Figure 6.** Illustration of (a) distributed stiffnesses and (b) the resulting stiffness being the coordinate transformed sum of the distributed ones

The analytical model employed is based on the model presented in Jaen-Sola (2017). It represents the quasi-static electromagnetic forces as classical springs distributed over the circumference, as shown in Fig. 6 (a). Each spring represents the resultant force over one of the $N$ sectors of the width $\beta$ (cf. Fig. 7 (a)). The different causes of unbalanced magnetic forces

are grouped in Jaen-Sola (2017) into so-called *modes*. The analytical equations, representing the stiffness per sector to model eccentricity $c_{PM}$, according to *mode 1*, are used here. The equations depend on the angle $\theta_i$, the sector width $\beta$, the mean eccentricity $\bar{\epsilon}$ and the eccentricity amplitude over the circumference $\hat{\epsilon}$ (cf. Fig. 7 (b)). An integration over $\beta$ is performed. Then, $\theta_i$ is a discretised variable running from $\frac{\beta}{2} : \beta : 2\pi - \frac{\beta}{2}$ to include each sector's spring force only once.

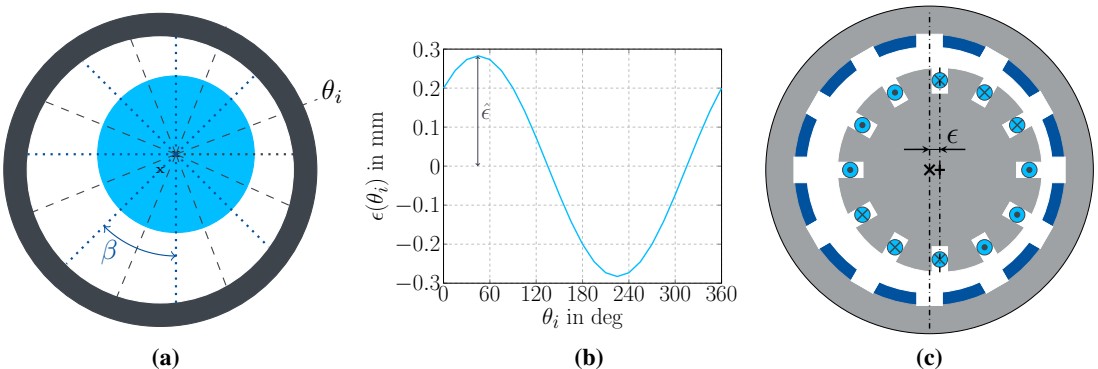

(a)                                  (b)                                  (c)

**Figure 7.** Parameter definitions of the analytical model (a) for the sector angle $\beta$ and the circumferential angle $\theta_i$, (b) the angle dependent eccentricity $\epsilon(\theta_i)$ and (c) the effective eccentricity $\epsilon$ as measured in the WT model. The position of the reference coordinate system for the outer rotor and bearings (**x**) is defined relative to the centre of the inner stator (**+**) with the distance $\epsilon$ in both transversal directions. Here, horizontal direction is shown as example.

The spring stiffness at each position of the circumference depends on the local, instantaneous air gap length $\delta(\theta_i)$ according

to Eq. 1. A mean eccentricity $\bar{\epsilon}$ would only occur, when changes of the structure's diameter due to thermal effects occur, which has not been investigated in this work. In the case of static eccentricity, the eccentricity amplitude $\hat{\epsilon}$ is constant. Such a static eccentricity is caused, e.g. during the component assembly, when the rotor is not perfectly centred relative to the stator. Static eccentricity has not been considered for this work. Dynamic eccentricity is characterised by a time-dependent eccentricity amplitude. The excitations of the wind turbine result in dynamically changing eccentricity amplitudes, which are provided

as input to the generator model. Therefore, this work focuses on dynamic eccentricity. The implementation into the Simpack model requires a resulting force in the direction of the shortest air gap only (compare Fig. 6 (b)).

$$\delta(\theta_i) = \delta_0 - \bar{\epsilon} - \hat{\epsilon}\sin(\theta_i) \tag{1}$$

To derive the resulting force from the local stiffnesses, the following procedure is used: The local sector stiffness is multiplied with the local air gap length. The resulting equation for local radial forces is evaluated for each sector $\theta_i$. The resulting local

forces are split into the global y and z components using cosine and sine of the local radial force. The forces are first summed up for the y and z directions and then combined to the resulting single radial force.

This force is the radial attraction force between rotor and stator, representing a single spring at the location of the shortest air gap, as in Fig. 6 (b). For an equally distributed air gap length, the force is expected to equal zero. Due to the finite discretisation of the generator into $N$ sectors, the resulting force does not equal exactly zero. Assuming one spring per pole the remaining

radial force is below $10^{-9}\frac{N}{m}$ which is considered sufficiently small for the application, and thus $N = 200$ is used.

The analytical function $F_{\text{emag}}(\epsilon)$ is derived and implemented as a force element at the generator location in parallel to the bearing force element. The electromagnetic forces are counteracting the bearing forces, which requires a reversed sign compared to the bearing, i.e. they can be seen as a spring with "negative stiffness". The eccentricity $\epsilon$ is measured dynamically during time integration as the radial distance between the generator rotor and stator centre (cf. Fig. 7 (c)). The resulting force

is applied to the generator rotor centre of gravity.

### 2.2.2  Numerical model

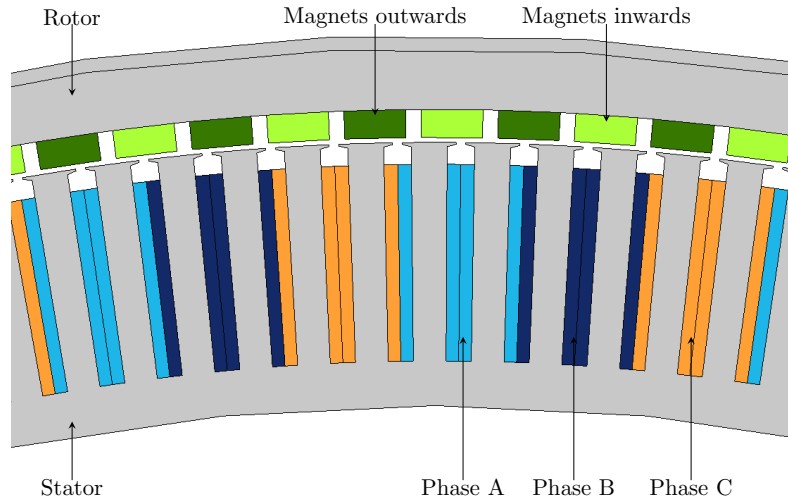

**Figure 8.** 2D cross-section of the numerical generator model with outer rotor and inner stator, showing the windings (light blue: phase A, dark blue: phase B and orange: phase C) and the magnets' orientation (dark green: outward magnets, light green: inward magnets). The winding layout follows the circular definition A' | A | A | A' | B | B' | B' | B | C' | C | C | C' | A | A' | A' | A | B' | B | B | B' | C | C' | C' | C.

For the numerical model, the generator was built in Comsol Multiphysics (Comsol) as a 2D-model. A part of the cross-section is shown in Fig. 8. The rotating machinery interface is chosen, solving Maxwell's equations based on a combination of the magnetic vector potential and magnetic scalar potential as the dependent variables. The stationary radial magnetic flux

density along the circumference of the air gap of the resulting generator model is given in Fig. 9 for rated conditions of rotational speed and torque. The magnetic flux density oscillates over the circumference. The maximum occurring magnetic flux density equals 1.2 T. For a theoretical, ideal machine, the oscillation is expected to be sinusoidal. Real machines differ from the sinusoidal oscillation due to the geometric shape of rotor and stator (cf. Fig. 9). The space dependent oscillations will change with the current over time. In consequence, the dynamic forces acting in the machine show overlaid high frequency

oscillations, which depend on the combination of the rotational speed, the frequency of the current, and the geometric shape of

the rotor and stator.

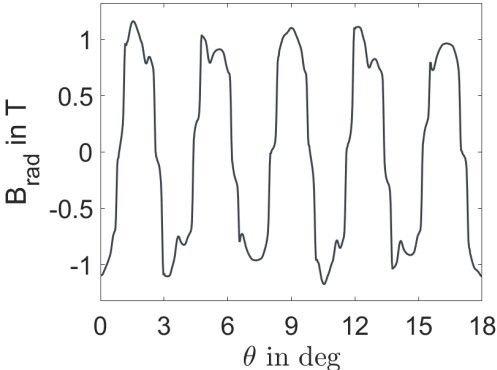

**Figure 9.** Stationary radial flux density $B_{\mathrm{rad}}$ over radial position $\theta$ along one pair of coils of the IEA 15 MW permanent magnet generator

The accuracy of the numerical model is achieved with a classical mesh convergence study. The model accuracy is quantified by the attraction forces in y and z direction, as these values are handed over to the mechanical WT model. Rotor and stator are kept aligned. Therefore, the expected solution for both forces is zero. The bending forces in the WT drive-train resulting out of the wind are in the magnitude of $100\,kN$ to $1\,MN$. Therefore, a remaining mesh error of up to $2\,\%$ is decided to be acceptable. The resulting mesh consists of 233,574 elements, giving a remaining attraction force in y direction of 1,021 N and in z direction of 2,192 N with a minimum element quality of 0.141, according to the skewness factor, and an average element quality of 0.7234.

For wind turbine applications, generators do not have a single operating point, as modern wind turbines operate at variable speeds and variable generator torque requirements. This cannot be captured by the analytical model, which is independent of the operating point defined by the winding current and the rotational speed, and depends only on the characteristics of the permanent magnets. In contrast, the numerical model can capture the influence of the operating point, as the impact of the winding current on the electromagnetic forces is included. In WTs, the winding current has to be adapted for each operating point below rated power to achieve the required generator torque. This dependency is included into the numerical model as a look-up table, and the required torque according to the WT controller is given as input from the MB model to the electro-magnetic model. The resulting fluctuations of rotational speed and current will add additional high-frequency oscillations to the forces. Based on the described set-up, a dynamic simulation for dynamic eccentricity and varying operating points can be conducted. The coupling of the FE model to the WT model is explained in details in Lüdecke et al. (2022). Simulating 15 s of fully coupled dynamic simulation with the numerical generator model and the wind turbine required about 14 days to be completed. The wind turbine model without generator or with analytical generator model requires about 1.5 to 4 hours for 650 s of simulation. Both simulations were set up using 4 cores on a machine with 512 GB RAM. Increasing the number of

cores does not speed up the simulation due to the limited size of the mesh of the numerical model, limiting the parallelisation capability of the solver. Nevertheless, it is expected that a first understanding of the interaction mechanisms can be obtained

based on the coupled simulations with the analytical model. Therefore, the numerical model is only used for validation of the analytical model and fully coupled simulations are omitted here.

### 2.2.3   Model comparison of analytical and numerical generator model

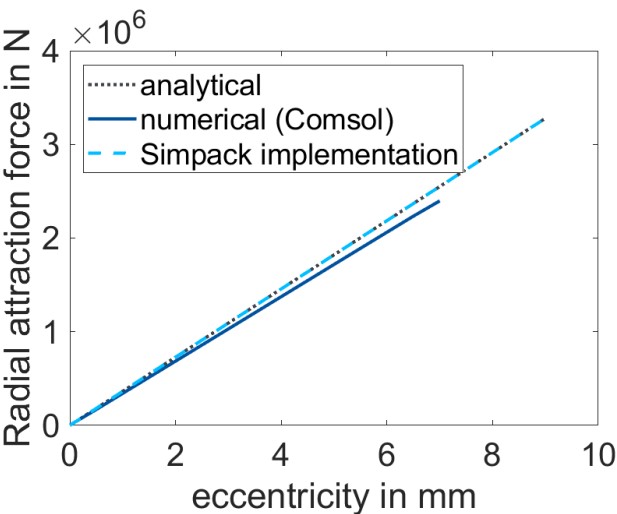

**Figure 10.** Comparison of analytical model implemented in Simpack with the numerical solution of Comsol for a static analysis that compares the dependency of effective eccentricity and radial attraction force for the analytical standalone model (gray dotted), the implementation of the analytical model in Simpack (light blue dashed), and the stationary solutions in Comsol (dark blue)

After the setup of the electromagnetic models, their implementation is compared. In the following, the relevant steps are explained. First, the stationary forces under eccentricity are analysed. Explicitly, the radial attraction force in the direction of

the smallest air gap for increasing eccentricities is calculated. For the analytical model, this means to evaluate the resulting equation for several eccentricities. To check the implementation of the derived equation into the Simpack model, the forces under constant eccentricity in steady state simulations are analysed. For the numerical simulations in Comsol, stationary simulations for each eccentricity at the rated operating point are performed. The comparison of the three solutions is given in Fig. 10 and shows a good agreement. The analytical solution from the Simpack implementation agrees exactly with the stand-alone

solution without WT model. Compared to the numerical solution, the analytical solution overestimates the attraction force and the absolute error is increasing with increasing eccentricity. Only for small eccentricities, the analytical model results in lower forces. This can be explained by the meshing error of the numerical model. Generally, the analytical solution can be considered a conservative estimation in comparison to the numerical solution. The error with reference to the numerical solution ranges between 6 % and 7.2 % with a mean error of 6.2 %.

In the second step, the implementation of the analytical model in Simpack was used to analyse the radial attraction force under

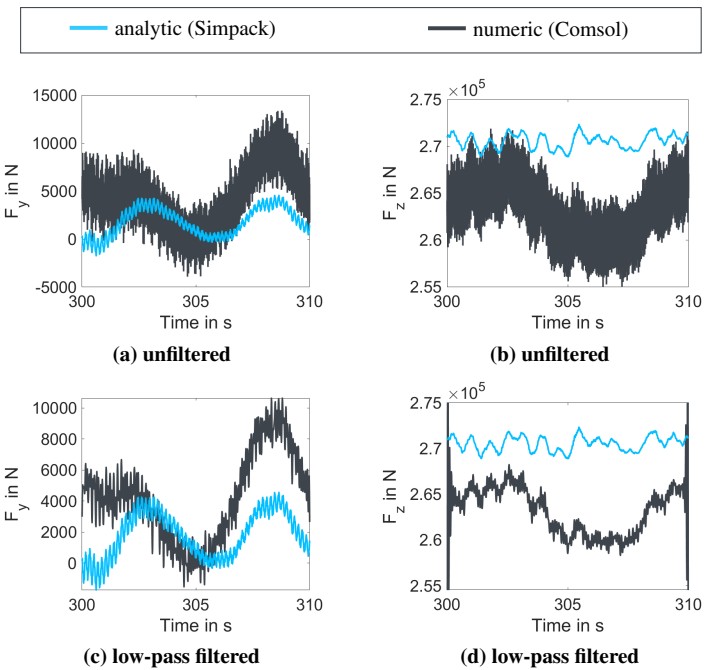

**Figure 11.** Comparison of analytical model implemented in Simpack with the numerical solution of Comsol for a dynamically changing eccentricity between the analytical model (light blue) and the numerical model (dark grey) for unfiltered numerical solutions (a) in y-direction and (b) in z-direction and low-pass filtered numerical solutions (c) in y-direction and (d) in z-direction with a passing frequency of 10 Hz.

dynamic loading, due to a turbulent wind field. The wind field has a mean wind speed of $10\,\frac{m}{s}$ and uses the NTM turbulence model with 5 % turbulence intensity. The calculated dynamic eccentricity, the demanded torque and the rotational angle of the rotor are given as input to the Comsol model and a dynamic simulation with variable winding current, rotational speed, and radial rotor position is performed to determine the numerical solution of the radial attraction force. Then the radial attraction

force from the Comsol and the Simpack solution are compared. The comparison is shown in Fig. 11. The upper two plots, (a) and (b), show the unfiltered numerical results compared to the analytical solution. The numerical solution contains high-frequency components. These high-frequency components can be explained with the physical effects outlined in Sect. 2.2.2. However, the major contribution to interactions is expected to result from low-frequency components. Therefore, the signals of the numerical solution are low-pass filtered with a cut-off frequency of 10 Hz. The comparison of the filtered numerical

results to the analytical result are given in Fig. 11 in the lower two plots (c) and (d). The filtering helps to better identify lower frequency fluctuations.

Generally, the eccentricity in z-direction is significantly higher than in y-direction, due to gravity. This corresponds with the two magnitudes higher attraction forces in z-direction than in y-direction. For both directions, the magnitudes of the forces

between analytical and numerical solution agree.

In y-direction, the analytical solution of the force remains smaller than the numerical solution, whereas in z-direction the opposite holds true. In combination with the knowledge about the remaining numerical error for zero eccentricity and the overestimation of the analytical model for high eccentricities (compare Fig. 10) the observed differences between the two directions can be explained. For small eccentricities the numerical error dominates the solution, whereas for higher eccentricities the overestimation of the analytical model dominates. This is also reflected in the mean differences of the compared signals. The mean difference in y-direction with respect to the numerical solution is -17.6 % and in z-direction is 2.8 %. Due to the lower magnitude of the force in y-direction, the comparably high difference can be tolerated. For z-direction, the difference is considered as sufficiently small to assume a good agreement.

Comparing the shape of the filtered numerical solution and the analytical solution, they match to a degree. The differences in Fig. 11 (c) and (d) are expected to result from the differences of the two models. The numerical model includes the influence of the geometric shape, rotational speed and demanded torque, which can not be captured by the analytical model. In addition, the numerical model includes a remaining error due to the meshing, which does not occur in the analytical model. The z-direction is showing two prominent frequencies in the numerical solution. The higher frequency with lower amplitude matches with the fluctuations visible in the analytical solution. The lower frequency, letting the numerical solution resemble one sinusoidal period, can not be directly explained. All in all, the comparison leads to the conclusion, that the two models deliver similar results and for higher eccentricities the analytical model is a conservative estimation compared to the numerical model.

The computational effort to simulate the coupled model of WT and numerical generator model is high, and only a few seconds of coupled simulation can be afforded with the available computational power. Nevertheless, based on the presented comparison of the two models it is assumed that coupled analysis based on the analytical model serves the scope which is to identify general mechanisms of electro-mechanical interactions on a WT level.

## 3 Electro-mechanical interactions in wind turbines

The present study aims to identify physical mechanisms that could lead to implications of electro-mechanical interactions to WT component loading outside the drive-train. To identify such interactions, the impact of the changed WT model on the system behaviour is analysed. Therefore, the structural dependencies are investigated in Sect. 3.1. Then, the feedback to the aerodynamic solver is examined in Sect. 3.2 and the controller's influence is evaluated in Sect. 3.3. Finally, Sect. 3.4 investigates how the identified interactions are influenced by the added DoF or the electromagnetic forces. A broader discussion of all identified interactions is outlined in Sect. 3.5. As test case of all these investigations, a periodic wind field with 1 % turbulence intensity and a mean wind speed of 8 m/s with 600 s of usable simulation time was used. However, the derivation of the natural frequencies of the coupled system is, in general, not dependent on the operating point. The same derivation could be done with simulations at different operating point or even with simpler simulation using e.g. static displacements as initial condition to identify the resonant frequency.

## 3.1 Structure

In the first step, the impact of the added DoF on the structural system characteristics is analysed. Therefore, the model system characteristics of the three simulation models according to Fig. 4 are compared and discussed.

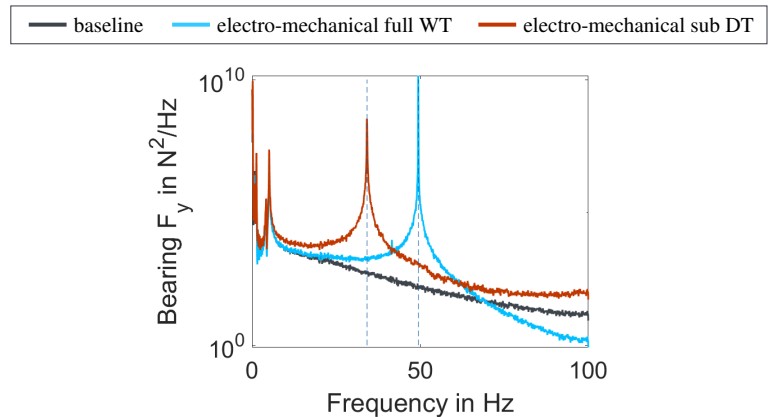

**Figure 12.** Comparison of spectra between the baseline model (grey) according to Fig. 4 (a), the drive-train (DT) substructure model (red) according to Fig. 4 (b) and the developed electro-mechanical model of the full WT (light blue) according to Fig. 4 (c) for the frequency range up to 100 Hz, based on an FFT of the main bearing load in global y-direction. Vertical lines mark the system mode, dominated by the main bearing moving in y-direction.

Fig. 12 compares the Fast Fourier Transformation (FFT) of the horizontal bearing force $F_{B,y}$. Differences only occur for a frequency above 20 Hz. The substructure and the full electro-mechanical model have an additional frequency that does not exist in the baseline model. It corresponds to the system mode of the bearings, introduced by the added DoF.

This frequency was estimated in Sect. 2.1 to 34.5 Hz based on a one-mass-spring system. In the substructure model the frequency equals 34.17 Hz and for the full electro-mechanical model increases to 49.5 Hz, a difference of 15.5 Hz.

The difference between the estimated frequency of Sect. 2.1 and the substructure model is expected to result from modelling uncertainties. The differences between the substructure model and the full electro-mechanical model result from coupled system modes that only occur when the tower and the drive-train are modelled in one system. Detailed explanations are given in the following, based on a simplified model:

Considering the position of the generator, the introduced bearing mode couples to two existing modes of the WT. Fig. 13

illustrates the position of the generator: In sub-figure (a) from the front and in (b) from the top. From these pictograms of the WT, it can be understood that the forces, acting in the horizontal direction in the generator, lead to a moment at the tower base around the x-axis (pointing downwind) and around the tower's vertical axis. Based on a model reduction, these DoFs can be simplified to a 2-DoF-2-mass system each, as given in Fig. 13 (c).

The spring with the stiffness $c_B$ represents the BS. The point masses are the non-rotating mass of tower and tower top $m_1$, and

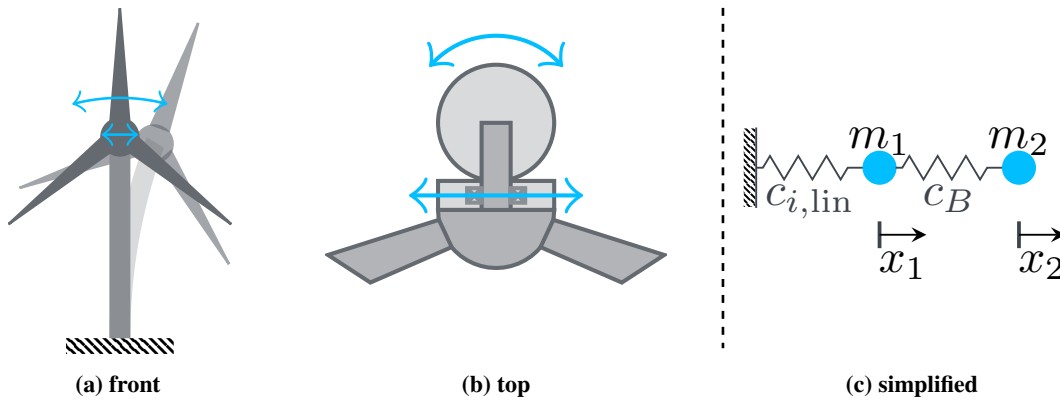

| (a) front | (b) top | (c) simplified |

**Figure 13.** WT pictograms looking at the turbine (a) from the front and (b) from the top, to illustrate the coupling of system modes between bearing and tower and (c) the derived simplified and linearised system of two masses and two springs representing tower and main bearing of a WT.

the rotating mass of WT rotor assembly and generator rotor $m_2$. The second spring $c_{i,\text{lin}}$ represents the tower stiffness. The rotational movements due to bending or torsion are expected to be small, keeping angles $\gamma$ below 5°. Therefore, $x_1$ can be described based on the linearised sine-function as $x_1 = \gamma \cdot l$. For the bending stiffness $c_{\text{SS,lin}}$, $l$ equals the hub height, while for the torsional stiffness $c_{\text{tor,lin}}$, $l$ equals the distance of the generator centre to the tower centre axis. The force acting on mass $m_1$ in the direction of $x_1$ on the example of torsion can then be written as $F_1 = c_{\text{tor}} \cdot \gamma = c_{\text{tor}} \cdot \frac{x_1}{l} = c_{\text{tor,lin}} \cdot x_1$ with $c_{\text{tor,lin}} = \frac{c_T}{l}$.

$$\begin{bmatrix} m_1 & 0 \\ 0 & m_2 \end{bmatrix} \cdot \begin{bmatrix} a_1 \\ a_2 \end{bmatrix} + \begin{bmatrix} c_{i,\text{lin}} + 2 \cdot c_B & -2 \cdot c_B \\ -2 \cdot c_B & 2 \cdot c_B \end{bmatrix} \cdot \begin{bmatrix} x_1 \\ x_2 \end{bmatrix} = 0 \qquad \text{with } i \,\epsilon\, [\text{SS, tor}] \qquad (2)$$

The system matrices of the reduced order models can be given in both cases with Equation 2. The solution of such a system is well known and returns two system modes. The first mode describes the in-phase oscillations of $m_1$ and $m_2$. It has the lower natural frequency, which is close to the frequencies of the first side-to-side or monopile torsional modes of the baseline model. The second mode describes the case of $m_1$ and $m_2$ moving in opposite directions. This mode has a natural frequency closer to $f_{\text{B,est}}$. In Sect. 2.1 the BS $c_B$ was estimated with 10 GN/m, and $m_2$ was given with 426.7 t. The stiffnesses $c_{\text{SS,lin}}$ and $c_{\text{tor,lin}}$ are estimated based on the baseline model and given in Table 3.

For the first system mode, the natural frequency based on the system parameters equals 0.19 Hz for side-to-side and 5.1 Hz for torsion, which equals the baseline model frequencies. For the second system mode, the natural frequency, in both cases, is 38 Hz, which is about 4 Hz higher than the one-mass-frequency that was calculated as $f_{\text{B,est}}$ in Sect. 2.1 and occurs in the isolated DT substructure model according to Fig. 4 (b).

This means, the coupling of the modes leads to an increase of the natural frequency. The 2-DoF-2-mass model results in a frequency, which is still about 10 Hz lower than the frequency of the full electro-mechanical model according to Fig. 4 (c).

**Table 3.** Model parameters from baseline model to determine the representative stiffness for a two DoFs representation and the determined values with a mass $(m_1 + m_2) = 2.446 \cdot 10^3 \, \text{t}$

| mode | $f_{\text{base}}$ in Hz | $c_{i,\text{lin}}$ in N/m |
|---|---|---|
| side-to-side | 0.190 | $3.48 \cdot 10^6$ |
| torsion | 5.138 | $2.55 \cdot 10^9$ |

The coupled system mode in the full electro-mechancial model couples the bending and torsion mode and includes the blades'
modes. These additional DoFs cause a further frequency increase of the coupled system mode at the bearings. Based on the presented results, the importance of this interaction mechanism can not be evaluated. It is expected that load and fatigue analyses are required to conclude about the relevance, which is beyond the scope of this work.

A first estimation of the impact of the chosen bearing stiffness $c_{\text{B}}$ on the identified system behaviour can be derived based on a sensitivity study of the 2-DoF-2-mass model. The resulting impact of the bearing stiffness on the system modes of tower
side-to-side bending, monopile torsion and coupled bearing system mode are shown in Fig. 14. The bearing stiffness only influences the coupled bearing system mode's frequency, but not the lower system frequencies of the tower. This underlines the made assumption that the identified electro-mechanical interaction mechanisms remain independent of the choice of the bearing stiffness.

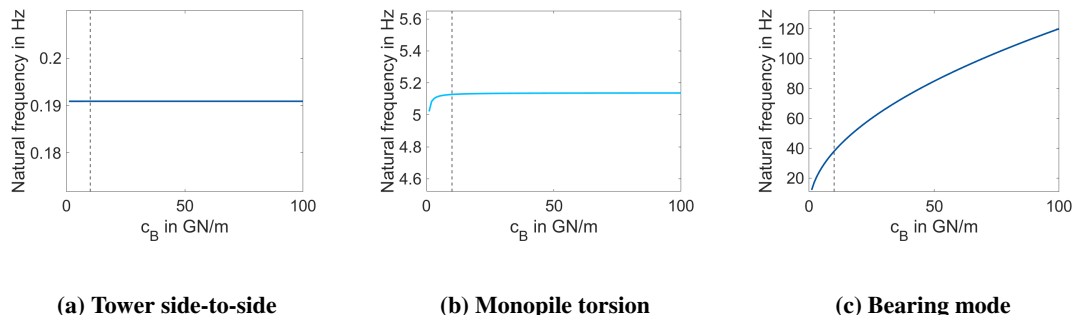

(a) Tower side-to-side      (b) Monopile torsion      (c) Bearing mode

**Figure 14.** Sensitivity analysis of the natural frequencies of the system modes of the 2-DoF-2-mass model according to Eq. 2 (a) tower side-to-side bending mode, (b) monopile torsional mode and (c) coupled bearing system mode depending on the chosen bearing stiffness $c_{\text{B}}$

In this study, only the radial DoF at the generator is taken into account. Other DoF that could be considered are the tilting
of the generator and axial displacements. Based on the approach of a simplified model as introduced in this section, these DoF could cause coupled modes with the fore-aft displacement of the turbine. As the fore-aft stiffness is expected to be close to the side-to-side stiffness, WTs with high bearing stiffnesses are likely to show similar impacts to the system modes. However, a detailed analysis is recommended for future work. In summary, the common two-step approach underestimates the natural frequency of the coupled bearing system mode compared to the one-step approach. This reveals the potential interaction mech-

anism of the main bearing and the electromagnetic generator forces with the WT structure.

## 3.2 Aerodynamics

With the enhanced understanding of the changes to the structural model, the next step is to look into possible interactions with the aerodynamic solver. Electro-mechanical interactions can only lead to interactions with the aerodynamics if the WT rotor is

390 affected in its position and velocity relative to the wind inflow. Due to the inertia of the flow, such interactions have a maximum frequency (Hansen, 2008, pp. 95). In the used aerodynamic solver of AeroDyn (Jonkman et al.) this is implemented as a low-pass filter to the module of unsteady aerodynamics. The cut-off frequency *filtCutOff*, has to be specified for each airfoil and is set to the default value of AeroDyn of 20 Hz for this work.

To investigate the interactions of the WT with the inflow, the aerodynamic forces calculated by the aerodynamic solver are

395 analysed. Specifically, the total aerodynamic force at the hub centre in y-direction of a rotating coordinate system is used for a spectral analysis. This force is directly acting in the radial direction of the bearing, affecting eccentricity.

As the interaction with the structure showed no impact on frequencies below 20 Hz, the aerodynamic solver is expected to not show any interactions. However, as mentioned in Sect. 2.1.2, the simulation results revealed a high sensitivity to the chosen communication time step, as shown in Fig. 15.

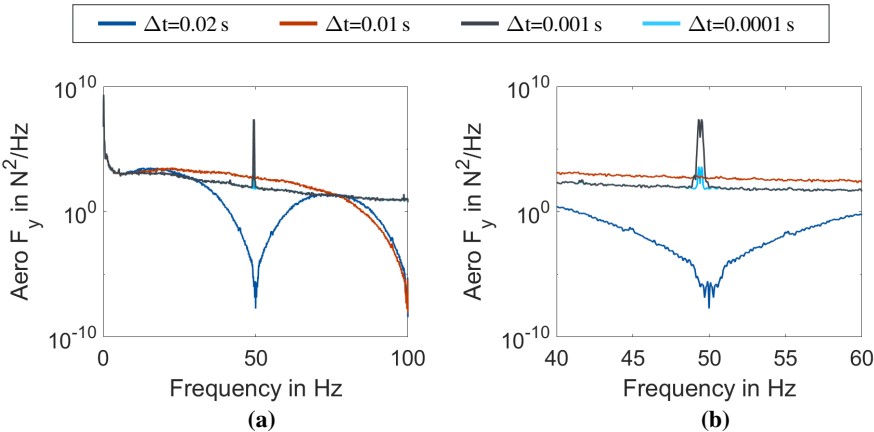

**Figure 15.** Comparison of frequency spectra of the total aerodynamic force in y-direction at the rotating hub centre from the aerodynamic solver in the frequency range (a) up to 100 Hz and (b) zoomed in to the range of 40 Hz to 60 Hz, for different communication time steps with the aerodynamic solver.

Below 20 Hz, all spectra are equal (cf. Fig. 15(a)), which corresponds to the results from the structural interaction. Above 20 Hz the communication time steps of 0.001 s shows a significant peak at the bearing modes' frequency. This peak is heavily reduced for 0.0001 s and does not occur for 0.01 s. For 0.02 s, the spectrum shows a dip around 50 Hz due to the communication

time step. The peak at the bearing mode's frequency contradicts the expected results. Additionally, a time series analysis reveals that the 0.02 s and 0.001 s communication time step lead to resonance like oscillations of the bearings, which do not occur for 0.01 s and 0.0001 s. According to Nyquist, the oscillations were to be expected for 0.02 s, but not for 0.001 s. The results indicate that the low-pass filter of the aerodynamic solver is not robust for such high frequencies, and the communication interval to the aerodynamic solver for electro-mechanical interactions has to be chosen carefully. Generally, interactions with the aerodynamics can be excluded for electro-mechanical WT models, though aerodynamic forces play a major role as system excitations.

## 3.3 Controller

In the next step, the impact of the controller to the system behaviour is investigated. The analysis is based on a comparison of open and closed loop simulation results. In the open loop simulation, the controller is switched off and constant rotational speed and blade pitch angle are set, according to the steady states of the turbine. The comparison of the open and closed loop simulation results helps to identify the impact of the controller to the turbine dynamics. This means, instabilities of the open loop model that are avoided by the control strategy can be identified, when oscillations with increasing amplitude occur in the open loop simulations, which can not be found in the closed loop simulations. Furthermore, oscillations occurring only in closed loop simulations are more likely to be caused by the controller, whereas oscillations that occur in open and closed loop simulations are caused by mechanical interactions, and oscillations only occurring in open loop simulations are successfully damped or compensated by the controller.

Running far off the normal operating points of the wind turbine can cause instabilities, in open loop simulations. Therefore, as test case, a periodic wind field with 1 % turbulence intensity and a mean wind speed of 8 m/s with 600 s of usable simulation time was used. The open loop model used a constant rotational speed of 5.7 rpm and zero degree pitch. These values were used as initial conditions for the closed loop model. In total, 650 s of simulation were evaluated, cutting off the first 50 s as numerical transients.

Fig. 16 (a) and (b) compare the tower top displacements in side-to-side direction for the baseline model and the electro-mechanical model in (a) for closed loop conditions and in (b) for open loop conditions. First, as the oscillation amplitudes of the open loop results do not increase over time, a mechanical instability can be excluded. Second, the amplitude of the closed loop simulation is higher for both simulation models. Generally, the results of baseline and electro-mechanical model in open loop (cf. subplot (b)) and closed loop (cf. subplot (a)) conditions are comparable, respectively.

An equivalent conclusion can be drawn from Fig. 16 (c) and (d), showing the time series of the tower top moment around its x-axis, pointing in downwind-direction. In the electro-mechanical model, higher frequency oscillations occur that do not occur in the baseline model. Looking further down the tower to the tower base (cf. Fig. 16 (e) and (f)), the tower's structural damping seems to decrease the level of higher frequency oscillations.

A quantification of the differences in loading to fatigue is omitted here, as the purpose of this work is, to identify interaction mechanisms. A clear interaction with the controller can not be identified. However, it is expected that the controller interaction is highly dependent on the BS parameter and the analysis has to be repeated for every new configuration.

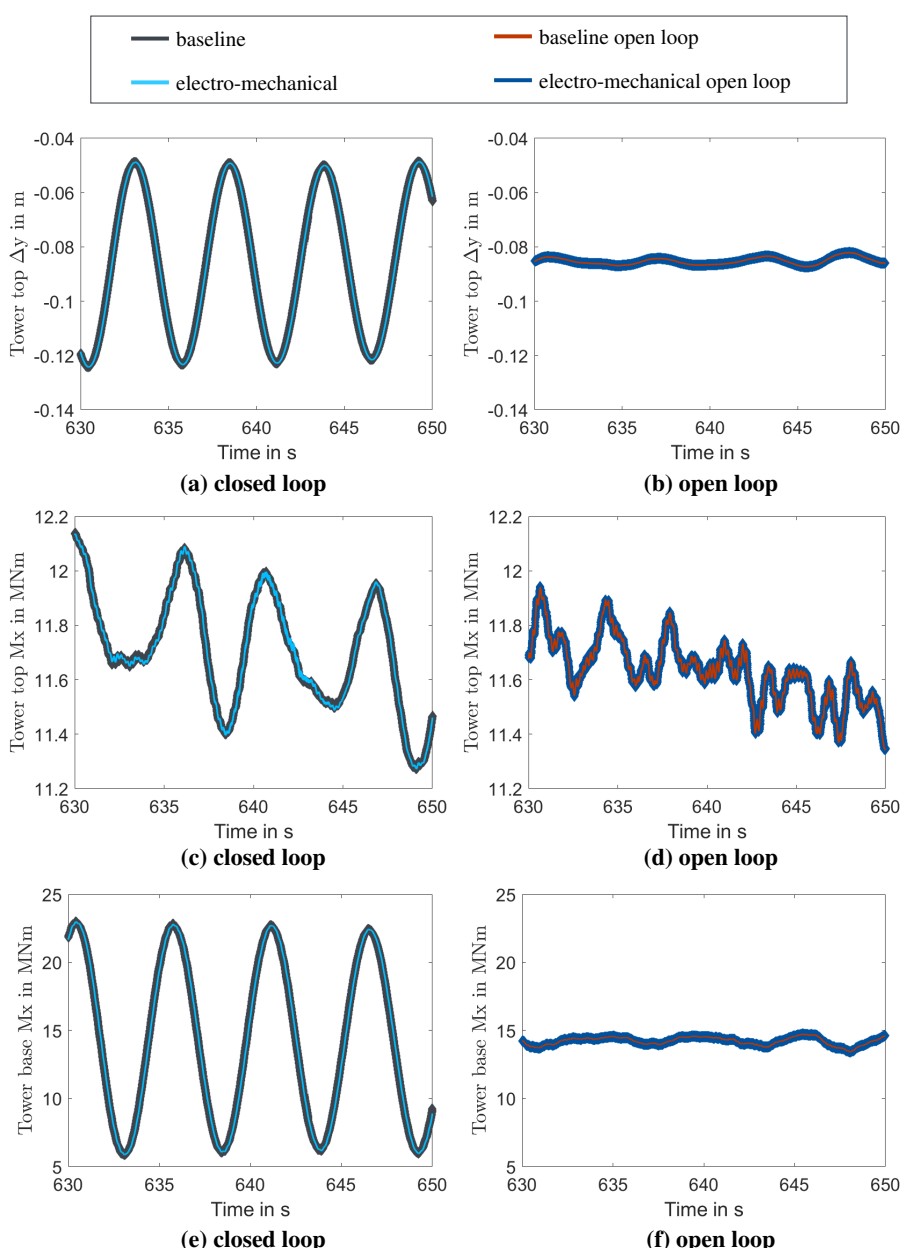

**Figure 16.** Comparison of tower behaviour in global y-direction with 8 m/s mean wind speed and a turbulence intensity of 1 % for tower top (TT) displacement in (a) closed loop simulation and (b) open loop simulation with constant rotational speed of 5.7 rpm, for tower top (TT) bending moment in (c) closed loop simulation and (d) open loop simulation and for tower base (TB) bending moment in (e) closed loop simulation and (f) open loop simulation

## 3.4 Impact of generator forces

The analysis of electro-mechanical interactions with the structure, the aerodynamic solver and the controller has shown that the system's natural frequencies can be impacted, when modelling electro-mechanical interactions. This section aims to differentiate between the impact of the added DoF and the influence of the electromagnetic generator forces interacting with the system.

Therefore, the full electro-mechanical model is compared to a model that includes the added DoF but omits the electromagnetic generator forces. Such conditions could occur, when the generator is switched off. Looking at the system modes, Fig. 17 shows a slight increase of the natural frequency of the bearing mode, when the generator is switched off. All other natural frequencies remain the same.

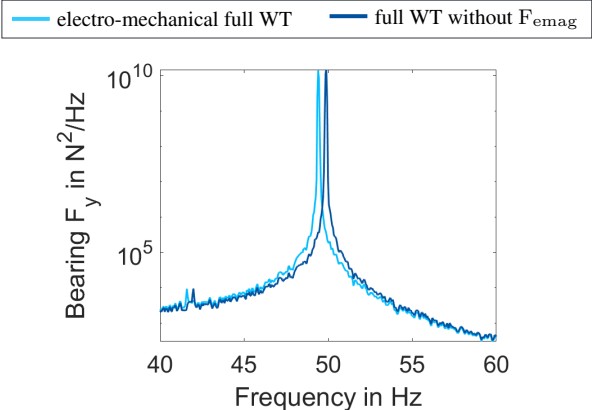

**Figure 17.** Comparison of spectra between the full electro-mechanical WT model (light blue) and the full WT model without electromagnetic generator forces ($F_{emag}$) (dark blue) for the frequency range of 40 Hz to 60 Hz, based on an FFT of the main bearing load in global y-direction.

The impact on the natural frequency can be explained by the characteristics of the electromagnetic field of the generator. The smaller the air gap between stator and rotor, the higher the attraction force. In consequence, the electromagnetic field acts self-exciting in case of eccentricity. From the point of view of the system, it can be seen as a non-linear spring with "negative stiffness", which acts in parallel to the bearing's stiffness $c_B$ and therefore, reduces the effective stiffness. As a result, the system's natural frequency of the bearing mode is reduced, when taking the electromagnetic generator forces into account.

The impact of the changed frequency on the simulation results is analysed based on the time series of the tower top moment in side-to-side direction, shown in Fig. 18. The comparison reveals that high-frequency oscillations in both models do not occur at the same time (cf. Fig. 18 (b)). These differences indicate a direct interaction of the electromagnetic generator forces with the WT system, though of low prominence.

In summary, the results indicate that the added DoF dominates the electro-mechanical interactions for the chosen system

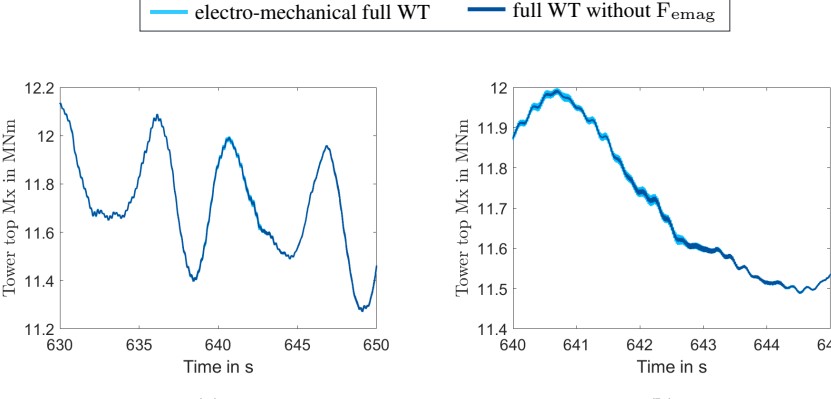

**Figure 18.** Comparison of time series of the tower top moment in side-to-side direction of the full electro-mechanical WT model (light blue) and the full WT model without electromagnetic generator forces ($F_{\mathrm{emag}}$) (dark blue) for (a) 20 s and (b) 5 s of a wind field with 8 m/s mean wind speed and a turbulence intensity of 1 %.

parameters. Nevertheless, the actual system response of occurring oscillations is impacted by the electromagnetic generator forces.

### 3.5    Discussion

The analysis of the interactions of the electro-mechanical generator model with the WT structure, the aerodynamic solver and the controller hints that electro-mechanical interactions can occur in WTs. The interactions result primarily from the additional DoF but are also influenced by the electromagnetic forces. The structural analysis shows a clear impact of the modelling approach (two-step vs. one-step) on the bearing's natural frequency. The results show that the DoF of the wind turbine com-

ponents interact, leading to coupled system modes, which only a fully coupled WT model can capture. Interactions with the aerodynamic solver and the controller were not identified for the investigated bearing stiffness. This presumably happens because both, the aerodynamic solver and the controller, apply low-pass filters to their input signals. The structural interactions, however, occur at frequencies above the cut-off frequency of the low-pass filters. Therefore, it is expected that interactions can also be excluded for other parameter combinations. Furthermore, the impact of the added DoF and the electromagnetic gener-

ator forces on the system's natural frequencies is expected to be a general phenomenon for all WT designs. In consequence, it is expected to have potential implications on the design procedure of WT components, e.g. bearings and generator and WT testing, specifically on nacelle testing.

**Impact on system design:** The design of main bearings, described in Hart et al. (2020), is dominated by the equivalent dynamic bearing load, which mainly depends on the dynamic radial bearing loads and the rotational speed (Schlecht, 2010, p. 198-209).

Therefore, changes of the frequency of the radial load oscillation can have an impact on the resulting bearing design load. Reliability analysis of operating wind turbines have shown that main bearings often do not reach their design lifetime (Hart

et al., 2019). The mismatch in system natural frequencies could potentially be a first hint for a reason. Nevertheless, a detailed load analysis is required to better understand the implications of the frequency shift to the bearing design load estimation.

Based on the 2-DoF-2-mass system of Sect. 3.1, other components of the drive-train can present a shift in their natural frequencies. Besides the bearing frequency, the natural frequencies of the generator structure, mounted to the wind turbine, are expected to differ from the design assumptions, causing a risk for system resonances. This expectation results from the common design approaches for generators, assuming a fixed support of the rotor for structural design, as explained in Tartt et al. (2021) and Jaen-Sola et al. (2019). It is outlined that design changes are used to adjust the component's natural frequencies according to the expected excitation frequencies under operation, i.e. rotational speed, and its harmonics: three and six due to the blades' passage in front of the tower, and the number of pole pairs due to interactions with the magnetic field. In Jaen-Sola et al. (2019), the support structure is stiffened compared to the optimised electro-mechanical design to increase the natural frequencies and avoid the resonance range of the operating conditions of the wind turbine. However, based on the results of this work, it has to be expected that the system natural frequencies of the mounted support structure are higher than those with fixed support. This could have two implications: First, as the assembled system would have a higher natural frequency than the isolated generator structure, the additional generator mass to stiffen the support structure might not be needed. Second, the frequencies of the structure with fixed support being below the operating range could increase and then potentially hit the operating range of the excitation frequencies, which would lead to resonances. To study the impact on the system frequencies and possible resonances in depth, a generator design with adapted support assumptions is required.

This component optimisation of the generator under consideration of dynamic wind turbine loads requires an iterative procedure (Jaen-Sola and McDonald, 2021), as the change of generator mass and mass moment of inertia impacts the resulting tower system frequencies. Moreover, when tower frequencies are changed, the dynamic loads considered in the optimisation are expected to change. In this iterative procedure, the additional change of the calculated dynamics due to the full electro-mechanical interactions could have implications, and it is suggested to be investigated in more details.

To reduce or avoid electro-mechanical interactions in radial direction Zhang and McDonald (2022) suggests splitting the stator windings into sectors and controlling the current sectorwise to balance the resulting radial forces. Assuming that the forces can be balanced at all times, this concept of operation would be equivalent to the analysis done in this work considering the switched-off generator. According to Sect. 3.4 this will increase the effective system frequency of the main bearings and may have implications on the occurring system oscillations. Therefore, this generator concept is suggested to be revisited to evaluate the potential benefits and drawbacks from a system design perspective. Possible resulting torsional coupling effects due to the impact on the resulting generator torque can not be discussed based on the results of this work but should also be considered in the evaluation of the concept to the system design.

The comparison of the analytical and numerical generator model has shown a high-frequency content in the electromagnetic forces of the numerical model resulting out of several design aspects, which are not captured by the quasi-static analytical model. Considering the increase of the bearing system frequency, there is potential for resonance-like interactions of the bearing frequency with the electromagnetic field. To study such interactions in detail, three options exist: the high-fidelity coupling of the numerical generator model and the wind turbine model has to be improved in terms of computational cost, a represen-

tative high-fidelity substructure model including the correct bearing frequency based on a scaled stiffness is derived, or the analytical generator model is extended to include dynamic transient behaviour of the electromagnetic field.

Following the trend of floating offshore wind turbines (FOWT), the simplified system could be extended by the flexibility of the wind turbine mooring system. In accordance with the results of this work, it has to be expected that this would further increase the natural frequency of the coupled system mode at the bearing, as the mooring system stiffness is expected to be comparably lower than the bearing stiffness. Sethuraman et al. (2017) analysed the impact of fully coupled electro-mechanical wind turbine models on the controller behaviour and concluded that a two-step approach is sufficient. The conclusion was based on the analysis of the controller signals being unaffected. As the controller, also in this work, does not show evidence for interactions but the coupled system modes affect the system behaviour, the conclusion has to be revisited.

**Impact on WT testing:** Besides the component design, the results could also have implications on nacelle testing on test benches. More specifically, such setups are not able to represent the coupled system modes, which may affect the meaning of the measured system dynamics. The technology review in Siddiqui et al. (2023) outlines that current nacelle testing is based on the assumptions that the system behaviour inside the drive-train does not feed back to the components outside the nacelle. In consequence, the setup equals a one-way coupling. The aerodynamic forces are often calculated based on simulations and applied to the tested nacelle at the hub connection point. Additionally, some include the tower top displacements. Considering both leads to a testing setup equivalent to Fig. 4 (b). According to the given results, this means, the testing setup is not able to capture the coupled system modes.

The purpose of nacelle testing, though, is to reduce the system testing costs in the design phase and at the same time ensure system reliability. Therefore, standardised tests are run, which cover normal operation and fault cases like grid loss. Especially, the fault cases are connected with highly dynamic system excitations and highly dynamic system reactions, e.g. from the controller. The excitation frequencies from the inflow, the controller dynamics and the electrical components remain the same with and without the coupled system modes. The dynamic behaviour of the drive-train components, though, can depend on the coupled system modes. In consequence, a significant offset of the structural, natural system frequencies could impact the final conclusion about the system reliability in these situations. Differences of the dynamics of nacelle components on a test bench or in the field have also been reported by Schkoda et al. (2016); Jassmann et al. (2021); Klein et al. (2023).

Generally, understanding the mechanisms of electro-mechanical interactions will, therefore, help to better optimise the system design and potentially reduce tower top mass. However, at the end, this depends on their impact on fatigue and ultimate loading, which will be investigated in future work. Such investigations should additionally consider to include structural damping effects, which were omitted in this work to reduce complexity. Higher frequencies usually come with lower amplitudes, which may be reduced further by damping. This could highly impact the conclusions of the load analysis.

**Computational effort:** The decision about the appropriate modelling fidelity for system design procedures, also, has to consider that the increase of the model fidelity comes at the cost of additional computational effort. In this regard, the communication time step with the aerodynamic solver is a bottleneck. If the time step could be increased from currently 0.0001s to 0.01s an overall simulation time of 1.5 h could be achieved which is about the computation time of the baseline model.

# 4 Conclusions

The presented work studies the effects of electro-mechanical forces on the system dynamics of WTs. Specifically, an analytical generator model from literature has been introduced to a multi-body model of the IEA 15 MW RWT. The analytical model has been validated against a high-fidelity FE-model of the generator and found to be in good agreement. The coupling to the WT model required an additional, radial DoF at the generator position, compared to the state-of-the-art WT model. Axial and tilting DoFs have been assumed to be rigid. The bearing model itself is simplified to a linear radial spring, omitting behaviour details, e.g. bearing pre-loading.

The model is used to identify changes to the structural system response due to the modelling approach. The results show that the introduced radial DoF at the generator creates coupled system modes with the tower that affect the system natural frequencies of the bearing mode. The electromagnetic generator forces cause a reduction of this natural frequency, as they behave as a spring with "negative stiffness" in parallel to the bearing stiffness.

The analysis of interactions with the aerodynamics reveals no interactions with the electromagnetic generator forces. Similarly, the analysis of interactions with the WT controller does not show evidence for interactions. Nevertheless, the communication time step for both, the aerodynamic solver and the controller, need to be adapted to the actual system design to avoid aliasing effects, causing unphysical resonance-like system behaviour.

Overall, in this work, the identified interactions are assumed to be design-independent general mechanisms. However, a parameter study investigating the impact of e.g. the BS is recommended to validate this assumption. Furthermore, the impact of the interaction mechanisms on the turbine fatigue loading has to be analysed in a next step to further evaluate the relevance of the identified interactions to the system design.

All in all, this work contributes to a profound physical understanding of the possible electro-mechanical interaction mechanisms. This prepares a future extension of the design space of WT generators based on a system design approach.

*Code availability.* The wind turbine model, set up in Simpack, is made available under DOI: 10.5281/zenodo.11099411.

*Author contributions.* FDL wrote the original draft. MS supported to set up the generator model. MS and PWC contributed to paper revisions.

*Competing interests.* No competing interestes are present.

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
