# Peer review of "Identification of electro-mechanical interactions in wind turbines"

_Wind Energy Science, 2024_

## Author Comment (AC1)

**Authors' Response (manuscript wes-2024-13)**

We thank all reviewers for their constructive comments that helped improve the quality of the manuscript and appreciate the time and effort they put into this. In the following pages, we reply to the comments on a point-by-point basis.

On behalf of the authors,

Fiona D. Lüdecke

**Answers to RC-1 (https://doi.org/10.5194/wes-2024-13-RC1)**

The authors would like to thank the reviewer for taking the time and providing useful feedback. All the comments have been taken into consideration and have contributed to improving the manuscript. A list of point-by-point replies to the comments follows:

| Reviewer's
comment | Is the main contribution related to eccentricity modelling when it comes to electrical-mechanical model? Then it should be clearer in the title or abstract.                                                                                                                                                                                                                                                                                                                                                                                                                                                                                                                                                                                                                                                                                                                                                                                                                                                                                                                                                                                                                                                                      |
|-----------------------|-----------------------------------------------------------------------------------------------------------------------------------------------------------------------------------------------------------------------------------------------------------------------------------------------------------------------------------------------------------------------------------------------------------------------------------------------------------------------------------------------------------------------------------------------------------------------------------------------------------------------------------------------------------------------------------------------------------------------------------------------------------------------------------------------------------------------------------------------------------------------------------------------------------------------------------------------------------------------------------------------------------------------------------------------------------------------------------------------------------------------------------------------------------------------------------------------------------------------------------|
| Answer                | The main contribution of the paper concerns the impact of modelling generator eccentricity to the system dynamics of the wind turbine. It aims to contribute to the research question, whether modelling generator eccentricity can be design driving for wind turbines. This is emphasised in the abstract in line 5 as follows: "This work aims to identify interactions of an additional degree of freedom in the radial direction of the generator with the wind turbine structure, the aerodynamics and the wind turbine controller." The electromagnetic model for forces resulting out of eccentricity is taken from literature and only the application to the wind turbine generator investigated here is validated. This is explained in the abstract as follows: "The analytical model, sourced from literature, is code-to-code validated against a finite element model of the generator in Comsol Multiphysics." The paper identifies electro-mechanical interactions of the generator and the wind turbine for radial generator eccentricity and discusses potential interaction with other degrees of freedom of the generator. Therefore, the authors would suggest to stick with the initially suggested title. |
| Changes to text       |                                                                                                                                                                                                                                                                                                                                                                                                                                                                                                                                                                                                                                                                                                                                                                                                                                                                                                                                                                                                                                                                                                                                                                                                                                   |

| Reviewer's
comment | There are different types of eccentricity - static, dynamic, and mixed. These exist also cases with axial misalignment. It seems that static eccentricity is the topic of the paper. Please elaborate.                                                                                                        |
|-----------------------|---------------------------------------------------------------------------------------------------------------------------------------------------------------------------------------------------------------------------------------------------------------------------------------------------------------|
| Answer                | In this work, purely dynamic eccentricity has been investigated. Static eccentricity would cause a constant force in the direction of the eccentricity, which is not expected to have a dynamic impact on the wind turbine behaviour. Axial misalignment has not been considered in the presented work. Axial |

|                 | misalignment would be equivalent to a mass imbalance, as often analysed for
blade imbalance at wind turbines. This force would be a rotating force caused by
the rotor mass rotating around the shaft axis out of centre and by the imbalance
of the electromagnetic forces. The analysis has shown that the generator radial
forces are two to three magnitudes smaller than the forces from the wind acting
on the shaft in cross-wind direction. Therefore, it is expected that the system
dynamics due to shaft misalignment will be dominated by the mechanical system
behaviour and electromagnetic forces will add up on these forces but not change
the overall dynamic response. The authors understand the necessity to better
clarify the assumptions about the considered eccentricity types and have added
a clarification about to the modelling section.                      |
|-----------------|----------------------------------------------------------------------------------------------------------------------------------------------------------------------------------------------------------------------------------------------------------------------------------------------------------------------------------------------------------------------------------------------------------------------------------------------------------------------------------------------------------------------------------------------------------------------------------------------------------------------------------------------------------------------------------------------------------------------------------------------------------------------------------------------------------------------------------------------------------------------------------------------------------------------------|
| Changes to text | Line 214:                                                                                                                                                                                                                                                                                                                                                                                                                                                                                                                                                                                                                                                                                                                                                                                                                                                                                                                  |
|                 | The spring stiffness at each position of the circumference depends on the local, instantaneous air gap length $\delta(\theta_i)$ according to Eq. 1. A mean eccentricity $\bar{\epsilon}$ would only occur, when changes of the structure's diameter due to thermal effects occur, which has not been investigated in this work. In the case of static eccentricity, the eccentricity amplitude $\hat{\epsilon}$ is constant. Such a static eccentricity is caused, e.g. during the component assembly, when the rotor is not perfectly centred relative to the stator. Static eccentricity has not been considered for this work. Dynamic eccentricity is characterised by a time-dependent eccentricity amplitude. The excitations of the wind turbine result in dynamically changing eccentricity amplitudes, which are provided as input to the generator model. Therefore, this work focuses on dynamic eccentricity. |
|                 | Line 262:                                                                                                                                                                                                                                                                                                                                                                                                                                                                                                                                                                                                                                                                                                                                                                                                                                                                                                                  |
|                 | Based on the described set-up, a dynamic simulation for dynamic eccentricity and varying operating points can be conducted.                                                                                                                                                                                                                                                                                                                                                                                                                                                                                                                                                                                                                                                                                                                                                                                                |
|                 | Line 286:                                                                                                                                                                                                                                                                                                                                                                                                                                                                                                                                                                                                                                                                                                                                                                                                                                                                                                                  |
|                 | The calculated dynamic eccentricity, the demanded torque and the rotational
angle of the rotor are given as input to the Comsol model and a dynamic
simulation with variable winding current, rotational speed, and radial rotor
position is performed to determine the numerical solution of the radial attraction
force.                                                                                                                                                                                                                                                                                                                                                                                                                                                                                                                                                                                     |

| Reviewer's
comment | More details regarding the generator characteristics should be included (e.g. pole and slot number).                                                        |
|-----------------------|-------------------------------------------------------------------------------------------------------------------------------------------------------------|
| Answer                | To better clarify the characteristics of the generator, a table is added to section 2.2 summarising the most important parameters of the IEA15MW generator. |
| Changes to text       | Table 2 is added in section 2.2                                                                                                                             |

| Line 196:                                                                        |
|----------------------------------------------------------------------------------|
| An overview of the outer-rotor generator design is given in Tab. 2. More details |
| about the definition of the generator can be found in Gaerther et al. (2020).    |

| Reviewer's
comment | Numerical model should be presented with more details as well. Is model 2D or 3D? Please include a cross-section of the model                                                                                                                                                                                                           |
|-----------------------|-----------------------------------------------------------------------------------------------------------------------------------------------------------------------------------------------------------------------------------------------------------------------------------------------------------------------------------------|
| Answer                | Additional information about the numerical modelling approach is added together with a cross-sectional view of one pole-pair, as the full cross-section overview does not allow to see any details due to the size of the generator.                                                                                                    |
| Changes to text       | Figure 8 was added showing the 2D cross-section and explaining the winding layout of the generator
Line 237:                                                                                                                                                                                                                         |
|                       | For the numerical model, the generator was built in Comsol Multiphysics (Comsol) as 2D-model. A part of the cross-section is shown in Fig. 8. The rotating machinery interface is chosen, solving Maxwell's equations based on a combination of the magnetic vector potential and magnetic scalar potential as the dependent variables. |

| Reviewer's
comment | Please show where z-axis and y-axis are. It was not clear for me. Normally (in electrical engineering literature) net force in eccentricity cases is characterized with Fx and Fy, assuming that airgap is uniformed in axial (z direction). It seems that in this paper the axis definitions are different. Please elaborate. |
|-----------------------|--------------------------------------------------------------------------------------------------------------------------------------------------------------------------------------------------------------------------------------------------------------------------------------------------------------------------------|
| Answer                | A figure is added, showing the coordinate definitions at different locations of the turbine, including the generator.                                                                                                                                                                                                          |
| Changes to text       | Figure 1 was added                                                                                                                                                                                                                                                                                                             |

**Answers to RC-2 (https://doi.org/10.5194/wes-2024-13-RC2)**

The authors would like to thank the reviewer for taking the time and providing useful feedback. All the comments have been taken into consideration and have contributed to improving the manuscript. A list of point-by-point replies to the comments follows:

| Reviewer's
comment | Line 15: Not true, especially onshore where MW/m^2 is decreased for low wind speed turbines                                                                                                                                                                                                                           |
|-----------------------|-----------------------------------------------------------------------------------------------------------------------------------------------------------------------------------------------------------------------------------------------------------------------------------------------------------------------|
| Answer                | The authors agree with the reviewer that in case of low wind speed sites the installed MW/m^2 is reduced. Sites with high wind speeds, though, show the tendency of increasing nominal power per wind turbine, which is connected with an increase of MW/m^2. As the clear differentiation between the cases requires |

|                 | a broader explanation in the paper but does not contribute directly to the content of the paper, the sentence in Line 5 was removed. |
|-----------------|--------------------------------------------------------------------------------------------------------------------------------------|
| Changes to text | Sentence removed                                                                                                                     |

| Reviewer's
comment | Line 88: A source or a figure would be good for a proof and how it was done                                                                                                                                                                                                                                                                                                                                                                                          |
|-----------------------|----------------------------------------------------------------------------------------------------------------------------------------------------------------------------------------------------------------------------------------------------------------------------------------------------------------------------------------------------------------------------------------------------------------------------------------------------------------------|
| Answer                | To outline the validation of the Simpack model against the OpenFAST model, two figures were added to the paper. The first one shows the comparison of steady states, whereas the second compares the dynamic behaviour of both models based on the system response to a stepped wind field. Additionally, the natural frequencies of the isolated components and the assembled system in Simpack were provided.                                                      |
| Changes to text       | Line 100:
The comparison of the steady states of both models are given in Fig. 2. From left to right, the power curve, the torque-speed curve and the pitch curve are shown. All the curves show a very good agreement. The comparison of the dynamic behaviour of the two models is based on a stepped wind field, which is shown in Fig. 3. This comparison confirms the agreement of both models under dynamic loading over the full operational range. |

| Reviewer's
comment | Line 130: How did you avoid overconstraining the system with two rigid axial constraints?                                                                                                                                                                                                                                                                                                                                                                                                                                                                                                                                                                                        |
|-----------------------|----------------------------------------------------------------------------------------------------------------------------------------------------------------------------------------------------------------------------------------------------------------------------------------------------------------------------------------------------------------------------------------------------------------------------------------------------------------------------------------------------------------------------------------------------------------------------------------------------------------------------------------------------------------------------------|
| Answer                | The shaft was modelled as a rigid body. Therefore, only one axial DoF needed to
be defined, which was placed at the centre of the generator. Over-constraining
the system would only be possible with a flexible shaft. The statement
"Therefore, the floating back bearing is reduced to a fixed bearing." in line 130 is
meant to explain that the defined force element only provides radial forces at
the location of the bearing as axial forces would not have an impact due to the
constraints in axial direction. To better clarify the chosen modelling approach to
the reader, a figure has been added together with a more detailed explanation. |
| Changes to text       | Figure 5 is added to the paper
Line 147:
A more detailed representation of the drive-train of the IEA15MWRWT is shown
in Fig. X. Subfigure (a) illustrates the position of the two bearings according to
the design in the report (Gaertner et al., 2020). The resulting implementation in
Simpack is shown in subfigure (b). Due to the assumption of a rigid shaft, one
axial constrain at the generator centre is sufficient and avoids over constraining
the system. In consequence, the bearings can be reduced to their supporting
behaviour, which is indicated by the radial springs in subfigure (b). Additionally,               |

| tilting of the generator is not included in this study, which is indicated by the |
|-----------------------------------------------------------------------------------|
| groove in the figure.                                                             |

| Reviewer's
comment | Line 139: Is the drivetrain tilt considered here?                                                                                                                                                                                                                                                            |
|-----------------------|--------------------------------------------------------------------------------------------------------------------------------------------------------------------------------------------------------------------------------------------------------------------------------------------------------------|
| Answer                | Yes, the drive-train tilt was considered. The configuration of the turbine is kept
the same as the IEA15MWRWT. Only the explained changes to include the
additional DoF have been performed. The general description of the turbine
model has been extended in subsection 2.1.1 for clarifications. |
| Changes to text       | Line 87:
It includes a drive-train tilt of 5 deg and blade cone angles of 2.5 deg in upwind
configuration.
Line 91:
Hydrodynamics around the monopile have not been considered.                                                                                                                  |

| Reviewer's
comment | Line 195: Please elaborate why no axial slicing was done and thus the effect of rotor tilting on the airgap was neglegted                                                                                                                                                                                                                                                                                                                                                                                                                                                                             |
|-----------------------|-------------------------------------------------------------------------------------------------------------------------------------------------------------------------------------------------------------------------------------------------------------------------------------------------------------------------------------------------------------------------------------------------------------------------------------------------------------------------------------------------------------------------------------------------------------------------------------------------------|
| Answer                | The authors agree to the reviewer that the tilting DoF should also be investigated to analyse electro-mechanical interactions in wind turbines. However, to ensure a clear focus of the paper, it was decided to reduce the number of DoF investigated in the paper and the radial DoF was chosen as a first step. To account for the necessity of analysing the omitted DoFs, potential implications of their impact based on the given results are discussed in subsection 3.1. Following the reviewers suggestion, an explanation about the choice to omit the tilting DoF was added to the paper. |
| Changes to text       | Line 64:
Based on the results in Duda et al. (2019) tilting is expected to influence occurring load levels. Potential impacts on the wind turbine dynamics, though, can not be accurately predetermined. However, combining several DoF will increase the complexity of the interactions. Therefore, it was decided to concentrate only on the radial DoF and its implications on the system response, to maintain clarity and coherence on the scope of the paper. Nevertheless, the extrapolation of the results in this work to the axial and tilting DoFs are discussed in Sect. 3.1.   |

| Reviewer's
comment | Line 239: Which loadcase was used here?                                                                                                                      |
|-----------------------|--------------------------------------------------------------------------------------------------------------------------------------------------------------|
| Answer                | The conducted simulation is based on a turbulent wind field and the resulting rotational speed and generator position depend on the dynamic behaviour of the |

|                 | wind turbine in normal operation based on the controller's choices. The details about the wind field have been added to the paper. |
|-----------------|------------------------------------------------------------------------------------------------------------------------------------|
| Changes to text | Line 285:                                                                                                                          |
|                 | The wind field has a mean wind speed of 10 m/s and uses the NTM turbulence model with 5 % turbulence intensity.                    |

| Reviewer's
comment | Line 325: Doesn't this rather reveal the necessity to consider ambient stiffnesses in WTs due to their complex dynamics? How does it change without the electromagnetic radial forces?                                                                                                                                                                                                                                                                                                                                                                                                                                                                                                                                                          |
|-----------------------|-------------------------------------------------------------------------------------------------------------------------------------------------------------------------------------------------------------------------------------------------------------------------------------------------------------------------------------------------------------------------------------------------------------------------------------------------------------------------------------------------------------------------------------------------------------------------------------------------------------------------------------------------------------------------------------------------------------------------------------------------|
| Answer                | In general, the generator forces reduce the effective stiffness of the bearings and
in consequence the system's natural frequency, as shown in section 3.4. The non-
linear behaviour of the generator forces makes the estimation of their impact to
the system dynamics and the interactions, identified in section 3.1, difficult and
requires further investigation. Even though the changes of the system behaviour
is dominated by the main bearing characteristics, the "negative stiffness" of the
generator forces is expected to influence the dynamics as seen from the results
in section 3.4. The authors agree with the reviewer that the given sentence may
be misleading and have adapted the sentence. |
| Changes to text       | Line 384:
This reveals the potential interaction mechanism of the main bearing and the electromagnetic generator forces with the WT structure.                                                                                                                                                                                                                                                                                                                                                                                                                                                                                                                                                                                        |

| Reviewer's
comment | Line 385: Which load case was chosen here? Were tangential emag loads considered in both models?                                                                                                                                                                                                                                                                                                                                                                                                                                                                                                                                                                                                                                                      |
|-----------------------|-------------------------------------------------------------------------------------------------------------------------------------------------------------------------------------------------------------------------------------------------------------------------------------------------------------------------------------------------------------------------------------------------------------------------------------------------------------------------------------------------------------------------------------------------------------------------------------------------------------------------------------------------------------------------------------------------------------------------------------------------------|
| Answer                | The question refers to the frequency analysis with and without electromagnetic forces. The simulation was the same as for section 3.1. The derivation of the natural frequencies of the coupled system is, in general, not dependent on the operating point. The same derivation could be done with simulations at different operating point or even with simpler simulation using e.g. static displacements as initial condition to identify the resonant frequency.                                                                                                                                                                                                                                                                                 |
|                       | Tangential electromagnetic forces were not included in the generator model, but
were based on the assumption of generator torque changing instantaneously
according to the controller commands in both models. In the implementation,
this implies the usage of a look-up table, updating the required generator torque
according to a given rotor speed. This assumption follows the state-of-the-art
approach, which assumes any changes of the generator torque to be too high in
frequency to be relevant for wind turbine load analysis. From the authors' point
of view, this is considered appropriate as torque changes would mainly lead to
electro-mechanical interactions in torsional direction of the shaft. For |

|                 | clarification to the reader, the modelling approach for generator torque has been outlined in section 2.2.                                                                                                                                                                                                                                                                                                                                                                                                                                                                          |
|-----------------|-------------------------------------------------------------------------------------------------------------------------------------------------------------------------------------------------------------------------------------------------------------------------------------------------------------------------------------------------------------------------------------------------------------------------------------------------------------------------------------------------------------------------------------------------------------------------------------|
| Changes to text | Line 198:
The interactions with the wind turbine focus on the radial generator forces.                                                                                                                                                                                                                                                                                                                                                                                                                                                                                    |
|                 | Variations of the generator torque due to torque ripple or due to the eccentricity are not considered. Those variations mainly apply to the torsional DoF, for which interactions have been investigated intensively in the literature, e.g. (Novakovic et al., 2013). Instead, the state-of-the-art approach is used for the generator torque, based on a look-up table of generator torque over rotational speed. Focusing on the radial forces only allows limiting the computational effort and better isolate the impact of the radial variations to the wind turbine loading. |

| Reviewer's
comment | General: How are rotorblades and tower modeled? What was the cut off frequency for eigenmodes?                                                                                                                                                                                                                                                                                                                                                                                                                                                                   |
|-----------------------|------------------------------------------------------------------------------------------------------------------------------------------------------------------------------------------------------------------------------------------------------------------------------------------------------------------------------------------------------------------------------------------------------------------------------------------------------------------------------------------------------------------------------------------------------------------|
| Answer                | Rotor blades and tower are modelled as flexible bodies in modal reduction using the first three modes of the blade (1.+2. Flap and 1. Edge) and the first four modes of the tower (1.+2. FA and 1.+2. SS). The substructure was also modelled as flexible body, based on the first 6 modes. In consequence, the highest system natural frequency of the wind turbine in baseline configuration equals 15 Hz. The introduction of section 2.1 is expanded, to include the modelling details and a table of the natural frequencies of the Simpack model is added. |
| Changes to text       | Line 94:
The modelling approach is equivalent to the implementation in OpenFAST, using flexible blades, tower and substructure. The resulting natural frequencies of the isolated components in one-sided clamping are provided in Tab. 1 together with those system natural frequencies of the coupled system, for which the according mode is predominant. The drive-train shaft has been modelled as a rigid component.                                                                                                                             |

| Reviewer's
comment | General: If the outcome was that the additional DOF is mainly responsible for
a change in the coupled dynamics, then the effect of the axial DOF on the WT
dynamics need to be studied as well                                                                                                                                                                                                                                                                                                                                                                            |
|-----------------------|---------------------------------------------------------------------------------------------------------------------------------------------------------------------------------------------------------------------------------------------------------------------------------------------------------------------------------------------------------------------------------------------------------------------------------------------------------------------------------------------------------------------------------------------------------------------------------|
| Answer                | The authors agree with the reviewer that from the perspective of the WT dynamics, the axial DoF may be of similar importance and should be investigated. Significant impacts of the generator forces to these interactions, though, are not expected, due to the limited impact of axial displacements to the radial generator forces. Additionally, the axial DoF would couple with the tower fore-aft mode, which is significantly higher damped through the aerodynamics than the side-to-side mode. It was decided to exclude the axial DoF from this study to keep a clear |

|                 | focus. However, a statement, mentioning the importance of investigating other DoF is mentioned in the paper at the end of section 3.1. |
|-----------------|----------------------------------------------------------------------------------------------------------------------------------------|
| Changes to text |                                                                                                                                        |

**Answers to RC-3 (https://doi.org/10.5194/wes-2024-13-RC3)**

The authors would like to thank the reviewer for taking the time and providing useful feedback. All the comments have been taken into consideration and have contributed to improving the manuscript. A list of point-by-point replies to the comments follows:

| Reviewer's
comment | However, the paper claimed that the Simpack model was validated against OpenFAST; no references or result comparisons are presented.                                                                                                                                                                                                                                                                                                                                         |
|-----------------------|------------------------------------------------------------------------------------------------------------------------------------------------------------------------------------------------------------------------------------------------------------------------------------------------------------------------------------------------------------------------------------------------------------------------------------------------------------------------------|
| Answer                | To outline the validation of the Simpack model against the OpenFAST model, two
figures were added to the paper. The first one shows the comparison of steady
states, whereas the second compares the dynamic behaviour of both models
based on the system response to a stepped wind field. Additionally, the natural
frequencies of the isolated components and the assembled system in Simpack
were provided.                                               |
| Changes to text       | Line 100:
The comparison of the steady states of both models are given in Fig. 2. From left
to right, the power curve, the torque-speed curve and the pitch curve are shown.
All the curves show a very good agreement. The comparison of the dynamic
behaviour of the two models is based on a stepped wind field, which is shown in
Fig. 3. This comparison confirms the agreement of both models under dynamic
loading over the full operational range. |

| Reviewer's
comment | It is not clear what the bearing arrangement is in this study. The paper just
mentioned the terms fixed and floating bearings without any more
information. More detail on the arrangement and specification of the bearings
is needed. |
|-----------------------|--------------------------------------------------------------------------------------------------------------------------------------------------------------------------------------------------------------------------------------------------|
| Answer                | A thorough description of the topology and the dimension of the drive-train taken from the original report (Gaertner et al., 2020) is given with the figure below:                                                                               |

|                 | $M_r^{z} \xrightarrow{M_r^{y}} M_r^{y}$ $F_r^{r} \xrightarrow{F_r^{y}} F_r^{x}$ $L_{hub}$ $W_r$                                                                                              | (a)                                                                                                                                                                                                                                         | (b)                                                                                                                                                                                                                                                                                                                                                           |
|-----------------|----------------------------------------------------------------------------------------------------------------------------------------------------------------------------------------------|---------------------------------------------------------------------------------------------------------------------------------------------------------------------------------------------------------------------------------------------|---------------------------------------------------------------------------------------------------------------------------------------------------------------------------------------------------------------------------------------------------------------------------------------------------------------------------------------------------------------|
|                 | Figure 5-2: CAD illust from Gaertner et al.,                                                                                                                                                 | tration of (a) the main sh
, (2020)                                                                                                                                                                                                      | aft and (b) turret (also called the nose); taken                                                                                                                                                                                                                                                                                                              |
|                 | This figure shows
with the location
these specification
report: "The pairs
downwind bearing
locating bearing
(Gaertner et al., 2                                           | s the configuration wit
s of the centre of gravi
ons in the figure the bea
ed set of bearings consi
ng. A tapered double
and a spherical roller
2020)                                                                     | h two bearings along the shaft together
ty of generator rotor and stator. Besides
aring types are specified as follows in the
sts of a fixed upwind bearing and floating
outer configuration was chosen for the
" bearing for the nonlocating bearing."                                                                                        |
|                 | A more detailed of
performed for the
been chosen to
additional details
which is out of the                                                                                       | drive-train design is not
nis work. Instead the r
avoid the necessity o
s would only be requi
ne scope of this work.                                                                                                            | included in the report and has not been
nodelling approach of the bearings has
of further details of the design. These
red for a load analysis of the bearings,                                                                                                                                                                                      |
|                 | To better clarify t
added together w                                                                                                                                                      | he chosen modelling a
with a more detailed ex                                                                                                                                                                                            | pproach to the reader, a figure has been
planation.                                                                                                                                                                                                                                                                                                        |
| Changes to text | Line 147:                                                                                                                                                                                    |                                                                                                                                                                                                                                             |                                                                                                                                                                                                                                                                                                                                                               |
|                 | A more detailed r
in Fig. 5. Subfigur
the design in the
the DoFs in Simp
shaft, one axial c
constraining the
supporting behav
Additionally, tilti
indicated by the t | epresentation of the di
re (a) illustrates the po
report Gaertner et al.,
ack is shown in subfigu
constraint at the gener
system. In consequence
viour, which is indicate
ing of the generator
track in the figure. | rive-train of the IEA 15MW RWT is shown
osition of the two bearings according to
(2020). The resulting implementation of
ure (b). Due to the assumption of a rigid
ator centre is sufficient and avoids over
ce, the bearings can be reduced to their
ed by the radial springs in subfigure (b).
is not included in this study, which is |

| Reviewer's
comment | A sensitivity analysis of the bearing stiffness against the response of the structure is needed.                                                                                                                                       |
|-----------------------|----------------------------------------------------------------------------------------------------------------------------------------------------------------------------------------------------------------------------------------|
| Answer                | The authors agree that investigations of the impact of different bearing stiffnesses on the dynamic WT response is required to evaluate the importance of the interactions to the WT design. Nevertheless, the authors expect that the |

|                 | identified interaction mechanisms are independent of the bearing stiffness. The bearing stiffness will mainly decide about the absolute natural frequency, but the coupling of modes will remain the same. To support this assumption a parameter study of the bearing stiffness based on the 2-DoF-2-mass system has been added to the paper. A detailed analysis of the impact of the bearing stiffness to the dynamic WT behaviour and its loading is proposed for future work as from the authors point of view it goes beyond the scope of this paper.                                                                                                         |
|-----------------|---------------------------------------------------------------------------------------------------------------------------------------------------------------------------------------------------------------------------------------------------------------------------------------------------------------------------------------------------------------------------------------------------------------------------------------------------------------------------------------------------------------------------------------------------------------------------------------------------------------------------------------------------------------------|
| Changes to text | Line 374:
A first estimation of the impact of the chosen bearing stiffness $c_{\rm B}$ on the identified system behaviour can be derived based on a sensitivity study of the 2-DoF-2-mass model. The resulting impact of the bearing stiffness on the system modes of tower side-to-side bending, monopile torsion and radial generator displacement are shown in Fig. 14. The bearing stiffness only influences the radial generator displacement frequency, but not the lower system frequencies. This underlines the made assumption that the identified electro-mechanical interaction mechanisms remain independent of the choice of the bearing stiffness. |

| Reviewer's
comment | Please consider the definition of the turbine and floater as a table in Wind Turbine Section 2.1. The coordinate system should be included too.                                                                                                                                                                                                                                                                                              |
|-----------------------|----------------------------------------------------------------------------------------------------------------------------------------------------------------------------------------------------------------------------------------------------------------------------------------------------------------------------------------------------------------------------------------------------------------------------------------------|
| Answer                | Following the reviewer's suggestion, a table of the natural frequencies of all
flexible components has been added to the paper. Additionally, the description
of the turbine model has been extended and a figure of the used coordinate
systems was added. Since this paper is based on the fixed bottom offshore wind
turbine with a monopile foundation, floater parameters have not been listed.                             |
| Changes to text       | Line 94:
The modelling approach is equivalent to the implementation in OpenFAST, using
flexible blades, tower and substructure. The resulting natural frequencies of the
isolated components in one-sided clamping are provided in Tab. 1 together with
those system natural frequencies of the coupled system, for which the according
mode is predominant. The drive-train shaft has been modelled as a rigid
component. |

| Reviewer's
comment | Please add the definition of the load cases in each section.                                                                                                                                                                                                                                                                                                                                                         |
|-----------------------|----------------------------------------------------------------------------------------------------------------------------------------------------------------------------------------------------------------------------------------------------------------------------------------------------------------------------------------------------------------------------------------------------------------------|
| Answer                | Most of the results in the paper show frequency analyses. The derivation of the natural frequencies of the coupled system is, in general, not dependent on the operating point. The same derivation could be done with simulations at different operating point or even with simpler simulation using e.g. static displacements as initial condition to identify the resonant frequency. Specifically, all frequency |

|                 | analysis were based on a wind field with 8 m/s mean wind speed and a turbulence intensity of 1 %.
Section 2.2.3, 3.3 and 3.4 also include time series results, which require the specification of the simulation setup. For all results the details about the used wind field have been stated.                                                                                                                                                                                                                  |
|-----------------|---------------------------------------------------------------------------------------------------------------------------------------------------------------------------------------------------------------------------------------------------------------------------------------------------------------------------------------------------------------------------------------------------------------------------------------------------------------------------------------------------------------------|
| Changes to text | Line 285:
The wind field has a mean wind speed of 10 m/s and uses the NTM turbulence
model with 5 % turbulence intensity.
Line 326:                                                                                                                                                                                                                                                                                                                                                                        |
|                 | As test case of all these investigations, a periodic wind field with 1 % turbulence intensity and a mean wind speed of 8 m/s with 600 s of usable simulation time was used. However, the derivation of the natural frequencies of the coupled system is, in general, not dependent on the operating point. The same derivation could be done with simulations at different operating point or even with simpler simulation using e.g. static displacements as initial condition to identify the resonant frequency. |

| Reviewer's
comment | It seems none of the cases are real DLCs in Section 3. In order to have a better
understanding of this added DOF, it's better to consider a DLC with NTM (wind)
and NSS (wave) and study the load on the generator or bearings.                                                                                                                                                                                                                                                                                                                      |
|-----------------------|------------------------------------------------------------------------------------------------------------------------------------------------------------------------------------------------------------------------------------------------------------------------------------------------------------------------------------------------------------------------------------------------------------------------------------------------------------------------------------------------------------------------------------------------------------|
| Answer                | The authors agree that the evaluation of the relevance of the interactions to the turbine design requires an in-depth investigation of the loads based on DLCs according to the standards and considering the damage equivalent loads. However, the scope of this work was to first identify the potential and characteristics of such interactions as a prior step to the load analysis. From the authors' point of view it is beneficial to keep the focus of the paper limited to ensure a clear storyline and keep the load analysis as the next step. |
| Changes to text       | -                                                                                                                                                                                                                                                                                                                                                                                                                                                                                                                                                          |

| Reviewer's
comment | The work is limited to a very specific condition in the generator. It is recommended to include this limitation in the title of the paper.                                                                                                                                                                                            |
|-----------------------|---------------------------------------------------------------------------------------------------------------------------------------------------------------------------------------------------------------------------------------------------------------------------------------------------------------------------------------|
| Answer                | The paper identifies electro-mechanical interactions of the generator and the wind turbine for radial generator eccentricity. However, it also discusses the implications for potential interaction with other degrees of freedom of the generator. Therefore, the authors would suggest to stick with the initially suggested title. |
| Changes to text       | -                                                                                                                                                                                                                                                                                                                                     |

| Reviewer's
comment | In Section 2.1.1, line 139, the maximum allowable radial eccentricity of the generator is assumed to be 2 mm according to the design. Please provide references for this assumption. Keeping a 2 mm radial distance for a 10 m generator needs reference. |
|-----------------------|-----------------------------------------------------------------------------------------------------------------------------------------------------------------------------------------------------------------------------------------------------------|
| Answer                | This assumption is taken from the report of the IEA15MWRWT generator (Gaertner et al., 2020). The reference is added to the sentence for clarification.                                                                                                   |
| Changes to text       | Line 159:
The maximum allowed radial eccentricity of the generator according to the design is 2 mm (Gaertner et al., 2020)                                                                                                                             |

| Reviewer's
comment | In Section 2.1.1, line 140, it is assumed that gravity loading should only cause a maximum of 10% of the allowed eccentricity. It is not clear how this assumption was validated, as this assumption is fundamental to the study. Further investigation in this regard or acquiring other references is needed.                                                                                                                                                                                                                                                                                                                |
|-----------------------|--------------------------------------------------------------------------------------------------------------------------------------------------------------------------------------------------------------------------------------------------------------------------------------------------------------------------------------------------------------------------------------------------------------------------------------------------------------------------------------------------------------------------------------------------------------------------------------------------------------------------------|
| Answer                | The authors agree that the bearing stiffness is key, when it comes to load analysis
of the wind turbine. Unfortunately, to the best knowledge of the authors,
definitions of equivalent bearing stiffnesses in wind turbines can not be found in
literature. That is why an assumption had to be made by the authors. However,
the absolute value of the bearing stiffness is not considered to impact the general
interaction mechanisms studied in this work. To better clarify how the value was
derived and what the expected impact of the choice is, the explanations in
section 2.1 were extended. |
| Changes to text       | Line 160:                                                                                                                                                                                                                                                                                                                                                                                                                                                                                                                                                                                                                      |
|                       | To the knowledge of the authors, no references about common bearing stiffnesses in wind turbines or eccentricity due to loading exist. Therefore, the assumption is made that the gravity loading should only cause a maximum of 10 % of the allowed eccentricity, i.e. 0.2 mm, the two bearings require an effective BS of 20.93 GN/m.                                                                                                                                                                                                                                                                                        |
|                       | Line 165:                                                                                                                                                                                                                                                                                                                                                                                                                                                                                                                                                                                                                      |
|                       | This value represents a first estimation for this study and needs further
investigation if a realistic load analysis is intended with the model. Nevertheless,
it will serve the purpose of analysing the interaction mechanisms, which are
expected to be independent of the exact value of the bearing stiffness.                                                                                                                                                                                                                                                                                                   |

| Reviewer's | In Section 2.1.1, line 141, it is assumed that the bearings have the same         |
|-------------------|-----------------------------------------------------------------------------------|
| comment           | stiffness. The internal and type of the bearings are not the same; therefore, the |
|                   | assumption is not correct.                                                        |

| Answer          | According to the report of the reference wind turbine a fixed front bearing and floating back bearing have been used for the drive-train design. Here, the front bearing carried radial and axial loads and the back bearing carries only radial loads. Due to the differences in the bearing types, it is likely that the bearing stiffnesses of are not equal.                                         |
|-----------------|----------------------------------------------------------------------------------------------------------------------------------------------------------------------------------------------------------------------------------------------------------------------------------------------------------------------------------------------------------------------------------------------------------|
|                 | Looking at the actual implementation of the DoFs in the wind turbine model, only
one effective bearing stiffness in radial direction is acting. This results out of the
decision to model the shaft as rigid body with the radial DoF, and no tilting DoF
and to reduce the bearings to the supporting forces acting like linear springs. This
aspect has been elaborated more in the paper. |
| Changes to text | Line 163:
Due to the modelling approach of a rigid shaft with only a radial DoF the
distribution of the effective bearing stiffness to the two equivalent force
elements does not influence the system behaviour. For simplicity, the bearing
stiffness is distributed equally to the two bearings.                                                                                          |

| Reviewer's
comment | In Section 2.2.2, line 222, it is stated that the coupling of the FEM and WT models proved to be too computationally expensive. It is better to present in a paragraph the quantitative cost of the FEM and WT models and compare the differences.                                                                                                                                                                                                                                                                                                                                                                                                                                                                                                                                                                                                                                                                                                                                                    |
|-----------------------|-------------------------------------------------------------------------------------------------------------------------------------------------------------------------------------------------------------------------------------------------------------------------------------------------------------------------------------------------------------------------------------------------------------------------------------------------------------------------------------------------------------------------------------------------------------------------------------------------------------------------------------------------------------------------------------------------------------------------------------------------------------------------------------------------------------------------------------------------------------------------------------------------------------------------------------------------------------------------------------------------------|
| Answer                | Following the suggestion of the reviewer, the computational costs of a fully coupled simulation have been outlined.                                                                                                                                                                                                                                                                                                                                                                                                                                                                                                                                                                                                                                                                                                                                                                                                                                                                                   |
| Changes to text       | Line 263:
The coupling of the FEM model to the WT model is explained in details in Lüdecke
et al. (2022). Simulating 15 s of fully coupled dynamic simulation with the
numerical generator model and the wind turbine required about 14 days to be
completed. The wind turbine model without generator or with analytical
generator model requires about 1.5 to 4 hours for 650 s of simulation. Both
simulations were set up using 4 cores on a machine with 512 GB RAM. Increasing
the number of cores does not speed up the simulation due to the limited size of
the mesh of the numerical model, limiting the parallelisation capability of the
solver. Nevertheless, it is expected that a first understanding of the interaction
mechanisms can be obtained based on the coupled simulations with the
analytical model. Therefore, the numerical model is only used for validation of
the analytical model and fully coupled simulations are omitted here. |

| Reviewer's | In Figure 10 (b, d, f), it is not easy to notice different graphs. However, the plots |
|-------------------|---------------------------------------------------------------------------------------|
| comment           | show the similarity of the curves; using markers could be helpful.                    |

| Answer          | Following the reviewers suggestion the figure was adapted to make both graphs visible. |
|-----------------|----------------------------------------------------------------------------------------|
| Changes to text | -                                                                                      |